Resource

# Assessing microbiome population dynamics using wild-type isogenic standardized hybrid (WISH)-tags

Benjamin B. J. Daniel ⓘ , Yves Steiger, Anna Sintsova, Christopher M. Field, Bidong D. Nguyen, Christopher Schubert ⓘ , Yassine Cherrak, Shinichi Sunagawa, Wolf-Dietrich Hardt & Julia A. Vorholt ⓘ ✉

Microbiomes feature recurrent compositional structures under given environmental conditions. However, these patterns may conceal diverse underlying population dynamics that require intrastrain resolution. Here we developed a genomic tagging system, termed wild-type isogenic standardized hybrid (WISH)-tags, that can be combined with quantitative polymerase chain reaction and next-generation sequencing for microbial strain enumeration. We experimentally validated the performance of 62 tags and showed that they can be differentiated with high precision. WISH-tags were introduced into model and non-model bacterial members of the mouse and plant microbiota. Intrastrain priority effects were tested using one species of isogenic barcoded bacteria in the murine gut and the *Arabidopsis* phyllosphere, both with and without microbiota context. We observed colonization resistance against late-arriving strains of *Salmonella* Typhimurium in the mouse gut, whereas the phyllosphere accommodated *Sphingomonas* latecomers in a manner proportional to their presence at the late inoculation timepoint. This demonstrates that WISH-tags are a resource for deciphering population dynamics underlying microbiome assembly across biological systems.

Microorganisms are ubiquitous in terrestrial and marine ecosystems. They form communities that exert a strong influence on these habitats, shaping them, and are in turn impacted by their environment. Microbiomes also contribute in various ways to the health and fitness of host organisms[1–3]. When microbes associate with multicellular organisms, for example, they form taxonomically structured communities indicating guiding principles that determine such outcomes and involve microbial interactions and microbe–microbe interactions[4,5].

Microorganisms impact each other during community assembly because of positive or negative interactions[6–14]. As a consequence, interactions may lead to priority effects. These effects, determined by the order of arrival of individual strains, play a crucial role in shaping community establishment[10,15–17]. However, the dynamics of microbiome assembly with high taxonomic resolution has remained difficult to assess, because it necessitates tracking individual lines of the same strain in a homogenous population over time. Genomic barcodes are ideal tools with which to address the current knowledge gap in community assembly by tracing population composition at an intrastrain resolution. Such tools have been applied most prominently to investigate population bottlenecks, with a special focus on pathogen infections[18–20]. They allow estimates of the severity of bottleneck events from endpoint measurements[21], and the impact of bottlenecks on establishing populations at a substrain level[22–24].

To ensure accurate capture of the population dynamics of isogenic bacterial strains, barcodes must be fitness neutral and are usually inserted at defined (neutral) sites[25]. This is in contrast to random

Institute of Microbiology, ETH Zurich, Zurich, Switzerland. ✉e-mail: jvorholt@ethz.ch

insertions with the aim of disrupting functions and quantifying fitness factors, which are reflected in a reduction (or increase) in the fraction of the total population (for example, using randomly barcoded transposon mutant library sequencing (RB-Tn-Seq))[26]. Barcodes should also avoid amplification biases. One way to achieve this is to create a set of defined barcodes using an empirically supported, standardized design and subsequent validation of performance. Together with the insertion of barcodes at neutral sites, this allows the generation of independently traceable isogenic strains that differ in the barcode. The application of independent tags is a common practice to mitigate the impact of spontaneous mutations. Barcoded strains were first developed for quantification by antibodies[27], by DNA-probe hybridization assays[28], then by quantitative polymerase chain reaction (qPCR)[21] and more recently by short-read sequencing[29–31]. Although qPCR permits a greater dynamic range and usually faster sample processing, next-generation sequencing (NGS) enables a more economical read out per sample when large numbers of barcodes or samples are assessed.

Gnotobiotic model systems are particularly well suited to test population outcomes of microbiomes because the input communities can be fully controlled[32]. Such systems have been introduced in the past for various biological systems. They comprise well-documented and representative strain collections that can be reconstituted to build synthetic microbial communities together with host systems. Both combined—host and microbes—enable the experimental testing of hypotheses by changing specific parameters or members of the community. A prominent example of microbiomes with established synthetic microbial communities is the mouse gut. Here, the Oligo-Mouse-Microbiota (OligoMM[12]) (ref. 5) has been introduced. For plants, *Arabidopsis thaliana* emerged as the most studied gnotobiotic system and the *At*-SPHERE has been assembled, which consists of collections of bacterial strains from roots and leaves, that recapitulates the majority of bacterial taxa found in environmentally grown heathy plants[33]. These strain collections from gut and plant are available through a public repository, making them accessible to the scientific community, and they have already contributed to our understanding of host–microbe interactions[34–37], as well as microbe–microbe interactions[11,12,38,39]. However, these experiments were often limited by the lack of a convenient, reproducible and cost-efficient system for assessing the underlying community dynamics in a quantitative fashion.

Here, we introduce a genomic barcoding system, termed wild-type isogenic standardized hybrid tags (WISH-tags) to mark model and non-model microbiota bacteria and test these in the presence and absence of synthetic communities from the mouse (OligoMM[12]) (ref. 5) and phyllosphere (*At*-LSPHERE)[11,33]. We apply barcoded focal strains to study intrastrain priority effects in the mouse gut and the phyllosphere of *A. thaliana*. We chose two biological systems that differ greatly in terms of strain mixing properties, presence of oxygen, nutrient availability, host tissue characteristics and development, and reveal differences in intrastrain behaviour between the two host systems. We provide a reliable and flexible tool that enables NGS and qPCR readouts, and demonstrate its applicability to different microbiomes. Thereby, we aim to contribute to the standardization of barcoding systems, which enhances interstudy comparability.

## Results

### Design and generation of the WISH-tags
To develop broadly applicable barcodes for the tracking of bacterial strains, we considered several factors concerning the versatility of their application as well as the sensitivity and specificity of their detection. First, we wanted to develop a system that allows the quantification of barcoded strain populations by both qPCR and NGS for more flexibility. On the one hand, qPCR offers an unparalleled dynamic range, allowing the quantification of abundant and rare barcodes from the same sample, while having a low turnaround time for small sample sizes. On the other hand, NGS-based quantification facilitates a higher throughput

of otherwise labour-intensive experiments, while being more economical for large sample numbers. Second, we wanted to ensure that the tags can be used for quantitative amplification over a large dynamic range, and third, that background amplification among the tags and genomic DNA of host model systems, here mouse and *Arabidopsis*, as well as their respective microbiota, was not detectable.

The WISH-tags developed in this study differed only in their core, the 40 bp of the unique barcode region, which ensures sufficient distinctiveness between any two tags (Supplementary Fig. 1). Within the unique barcode region lies the binding site for the unique reverse primer for qPCR, which together with the universal forward primer (also used for NGS) enables the quantification and identification of a given WISH-tag by means of qPCR (Fig. 1a). The resulting amplicon is 88 bp long, which is within the optimal length for qPCR products[40], whereas the primer sites for the NGS amplicon are spaced to produce a fragment of 120 bp in addition to the length of the overhangs, which is dependent on the indexing method used. The resulting amplicon is ideally suited for paired-end sequencing for high accuracy, because the unique barcodes are sequenced in both directions, helping to mitigate the impact of sequencing errors.

To generate sequences for the WISH-tags, we developed an adaptable workflow (Fig. 1b,c and Methods). The individual primers and spacers were generated separately by creating random strings of bases. These sequences were then filtered to ensure a balanced GC content and exclude palindromic sequences. Following the filtering steps, the primers had to meet extra conditions. Specifically, primers were not allowed to match genome sequences on the exclusion list (tolerance of up to two mismatches) based on our intended experimental host microbiota systems (the host organisms *Mus musculus* and *Arabidopsis thaliana*), as well as the genomes of microbiota collections[5,33]. Sequences with four or more identical bases or consecutive duplets of bases were also removed and finally Primer3 was used to test for uniform melting temperatures[41], resulting in a total of 123 billion possible barcodes. The different primers and spacers were then combined to form complete WISH-tags.

### Validation of WISH-tag amplification using qPCR and NGS
To verify the performance of the WISH-tags, we assessed them using qPCR and NGS. To ensure orthogonality, 62 WISH-tags were tested experimentally against each other and in combination with pooled DNA from faecal pellets of mice raised in a germ-free environment, axenic *A. thaliana* Col0 plants and their respective microbiota collections. We refer to this pooled DNA as bacteria, plant and mouse DNA (BPM-DNA).

Initially, we mixed WISH-tags and amplified these using qPCR. The resulting signal surpassed the background noise by five orders of magnitude (Fig. 2a). The addition of BPM-DNA (more than 3,300-fold excess) did not compromise the amplification (Fig. 2a). To test specificity and potential off-target amplification, we performed dropout experiments by leaving out one tag at a time, thus creating 62 different mixes, and adding BPM-DNA. These controls, performed individually, confirmed that each tag was amplified specifically because signals were below the detection limit, with three producing a signal just above it (Fig. 2a and Supplementary Fig. 2). These results indicate that unique primers do not bind to mismatched barcode sequences, demonstrating the high specificity of the WISH-tags for qPCR. Next, we evaluated all 62 WISH-tags by NGS, for which they were combined into mixes of eight WISH-tags of equal concentration. Except for two outliers, the separation between background noise and signal was again five orders of magnitude and resulted in consistent detection of the tags in terms of number of counts, indicating that they perform equally well (Fig. 2b).

After validating the specificity of the tags, we probed whether the amplification was linear. To this end, we created a range of serial eightfold dilutions using the same groups of eight tags each to ensure that each mix (Mix 1–8) was represented once at each dilution across the eight libraries (Libraries A–H) that were prepared for this benchmark (Extended Data Fig. 1a,b). The expected number of copies for

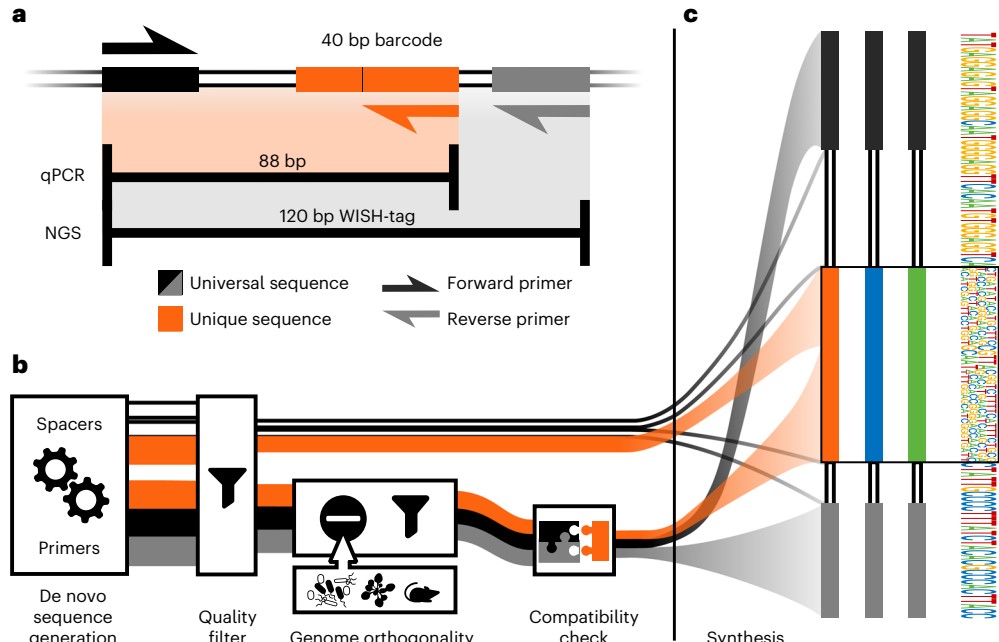

**Fig. 1 | Design and principles of WISH-tag construction. a**, WISH-tags consist of several parts that allow specific qPCR amplification and identification of a central barcode when using NGS. The 40 bp barcode includes the unique reverse primer (24 bp), which together with the universal forward primer (24 bp) enables amplicon sequencing of an 88 bp product that also contains a universal spacer. Amplification for NGS is achieved by the same universal forward primer together with a universal reverse primer, resulting in a 120 bp product that contains two universal spacer regions. **b**, De novo sequence generation pipeline. The candidate sequences for the parts were generated as random compositions of the four bases A, C, G and T. These were subsequently filtered for non-palindromic sequences using a balanced GC content. Primer sequences containing runs of four or more identical bases or consecutive duplets were removed. Sequences that aligned with up to two mismatches against the background genomes (mouse, plant, microbiota) were also removed and Primer3 (ref. 41) was applied to ensure similar melting temperatures. The universal primers were fixed for all WISH-tags, whereas the unique primers were constructed individually for each one. The thickness of the lines in the flow chart represents the number of sequences being processed (not to scale). **c**, The first 500 WISH-tags were used to generate a sequence motif illustrating the frequency of each base along its entire length.

each barcode matched the theoretical expectations over five orders of magnitude (Fig. 2c coloured loess regression lines[42] versus black line; Extended Data Fig. 1c and Fig. 2d) down to 100 reads, showing that the tags can be used reliably in a large range of dilutions to quantify tags.

### Strain tagging and validation of barcoded strains

Pursuing our overarching goal of creating a barcoding system that can be applied in distinct genomically tractable organisms, we selected ten strains to integrate a set of WISH-tags in each. Because the WISH-tags are independent of the integration method, we choose the most readily available one for each strain. Three *Salmonella enterica* serovar Typhimurium (*S.* Tm) strains were barcoded using the λ-red system in combination with pSIM for site-directed integration[43,44]. These strains were ATCC14028S[45] and SL1344 (SB300)[46], which are commonly used in the field of enteric bacteria research, as well as an avirulent mutant of SL1344 with deletions in Δ*invG* and Δ*ssaV*. We also barcoded the mouse gut commensal *Escherichia coli* 8178 (ref. 47). In addition, we tagged six representative species of the *At*-LSPHERE[33] with six barcodes each; the barcodes were unique to the species. These were *Sphingomonas* Leaf257, *Duganella* Leaf61, *Methylobacterium* Leaf88, *Xanthomonas* Leaf131, *Xylophilus* Leaf220 and *Rhizobium* Leaf68. Strains were barcoded with WISH-tags using a Tn7-based approach for integration downstream of the *glmS* gene[48] (Methods). The genome of *Rhizobium* Leaf68 could not be targeted by the Tn7 transposase; we therefore used homologous recombination for the integration at the *glmS* site. In total, more than 60 WISH-tags were integrated across the ten strains (Supplementary Table 1). The integration of tags was verified by PCR and the strains were subsequently assessed for fitness in vitro or in vivo, here, mouse and plant colonization (Supplementary Figs. 3–5). Overall, our results revealed that the barcodes had no detectable fitness impact on the bacteria.

### Dynamics of a commensal-like *S.* Tm mutant in the murine gut

To demonstrate the potential of WISH-tags, we conducted two proof-of-concept experiments on the exploration of intrastrain priority effects. The first experimental system was the mouse gut as it is one of the most well-studied microbial habitats and serves as an accessible experimental model system for the mammalian gut. The gut is unique in the way in which environmental factors like temperature, pH and nutrient availability are host controlled. Previous research indicated that stronger priority effects are observed among more closely related strains during gut colonization, as revealed by whole metagenomic sequencing[17]. Evidently, isogenic strains feature the highest potential niche occupation possible for an incoming strain, thus providing a test case for the application of WISH-tags to distinguish intrastrain interaction outcomes and priority effects.

The experimental approach was to separately introduce two sets of isogenic strains into the microbiota: an 'early arrival' group and a 'late arrival' group. As the focal strain for these experiments, we chose an avirulent mutant of *Salmonella* Typhimurium SB300 Δ*invG* Δ*ssaV*[49] to avoid disease progression during the experiment. This strain can colonize the gut lumen but does not invade the gut tissue and cause mucosal disease[50]. Therefore, it is ideally suited to probe microbial growth and competition in the animal gut lumen during the initial growth phase of an infection, when the pathogen has to grow within an undisturbed gut environment[35,49]. The strain was barcoded with six different WISH-tags, creating unique, traceable isogenic populations (lines). We then split the six lines of this focal strain into two sets of three, allowing us to independently validate the outcomes recorded with one set of strains. The early arrivals were inoculated as a 1:1:1 mixture of three uniquely WISH-tagged strains (5 × 10⁷ colony-forming units (CFU), by oral gavage) and allowed 4 days to establish and reach the respective carrying

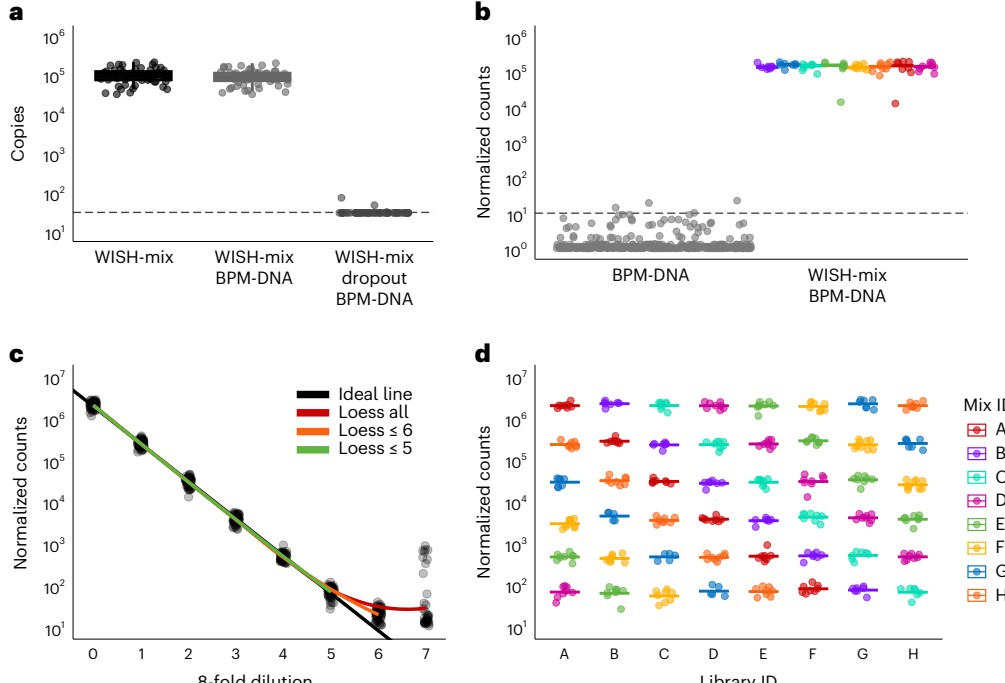

**Fig. 2 | Validation of the WISH-tag amplification. a**, Evaluation of the specificity of WISH-tags by qPCR when using a mix of 62 WISH-tags. The *y* axis shows the copy number determined for each tag and the *x* axis indicates the tested condition. In the WISH-mix condition, all tags were added in equal concentrations (~100,000 per tag). The condition WISH-mix BPM-DNA contained genomic DNA from bacteria, plant and mouse (referred to as BPM-DNA) added to the WISH-mix. The last condition, WISH-mix dropout BPM-DNA, displays the signal for each individual WISH-tag if that specific tag was excluded from the entire mix while BPM-DNA was present. The dashed line represents the limit of detection, set at a cycle threshold value of 32. The centre line represents the median, with the upper and lower edges of the box representing the 25th and 75th percentiles, respectively. The whiskers extend at most 1.5 × the interquartile range (i.q.r.). The data broken down by barcode are given in Supplementary Fig. 2. **b**, Evaluation of the specificity of WISH-tags by NGS. The test was performed in the presence of an excess of BPM-DNA (0.003 ng per WISH-tag to 10 ng of

BPM-DNA for a 3,333-fold excess). Reads obtained solely from BPM-DNA are displayed on the left, and reads obtained when eight mixtures, each containing eight WISH-tags at an even concentration, were added are shown on the right. Colours in the visualization represent the Mix_ID. Each data point represents the total reads obtained for a specific WISH-tag, and the bar represents the mean of the normalized counts in a mix. **c**, Assessment of linearity of the NGS readout across dilutions. Normalized counts returned for each WISH-tag across eight 8-fold dilution steps are shown. The black line represents the theoretical signal, and loess models are drawn on top using coloured lines[42]. The red model takes all points into account, the orange model considers points up to dilution 6 and the green model considers points up to dilution 5. **d**, The data from **c** are presented here up to dilution 5, with the colour indicating the Mix_ID and the *x* axis representing the Library ID. Each bar represents the mean of the normalized counts in a mix (the number of barcodes in each mix is $n = 8$, except for mix B and G, where $n = 7$).

capacity within the mouse gut[5,51]. Late arrivals were introduced 4 days later (1:1:1 mixture of strains) and given an equivalent amount of time to establish (Fig. 3a; see Methods and Extended Data Fig. 2a for details and mock controls). Faecal pellets were collected daily for microbiota assessment. We sampled faeces because earlier data had established that faecal *S*. Tm populations recapitulate the *S*. Tm population in the caecum[19], which is the main site of gut-luminal *S*. Tm growth in germ-free (GF) and OligoMM[12] mice. The experiment was conducted twice, with each set of three isogenic strains serving once as early arrivals and once as late arrivals to cross-validate the data and provide an independent biological replicate. We performed the experiments in GF and OligoMM[12] mice to test the effect of the presence of other commensal microbes on the colonization dynamics of the focal strain. Mice were bred in isolators and transferred into individually ventilated cages so that the colonization experiments were performed under strict hygiene barrier conditions. Overall, the experiment included a total of six different variations of our set-up, as depicted in Extended Data Fig. 2a. Each was tested in two independent experiments involving a total of six mice per treatment type. The barcodes were quantified by Illumina sequencing.

In GF mice, the gut was rapidly colonized by the focal strain, which reached carrying capacity within a single day (Fig. 3b). The population attained carrying capacity at ~10[10] CFU per g of faeces, as expected[51]. Colonization of the focal strain in the OligoMM[12] mice increased

gradually and reached 10[8] CFU g[−1] after ~2 days, in line with previous observations[5]. These population levels were reached irrespective of the inoculation timepoint, as revealed from control treatments in which inoculation was performed only at either one of the timepoints (Fig. 3b and Supplementary Fig. 6). By monitoring lipocalin-2 concentrations during the experiment as a marker for intestinal inflammation, we verified that the avirulent *S*. Tm strains used in our experiment did not cause overt inflammation in mice (as indicated by lipocalin-2 values of <10[3] ng per g of faeces; Supplementary Fig.7).

We assessed priority effects by comparing the early-arriving and late-arriving strains. In the GF mouse experiment, the early arrival population was ~10[10] *S*. Tm cells on day 4, whereas the late arrivals, inoculated on day 4, comprised 5 × 10[7] *S*. Tm cells (Methods). In the case of simple intermixing, we should have detected late arrivals in every single mouse between days 5 and 8 (Fig. 3b, upper left) because our detection limit for late arrivals by WISH-tag sequence counting was ~2 × 10[6] (dashed line in Fig. 3b). However, our quantitative analysis revealed much lower densities of the late arrivals than expected (Fig. 3b, upper left). This is consistent with a pronounced priority effect. Similar considerations apply to the OligoMM[12] mouse conditions (10[8] total population size for the early arrivals at 4 days post inoculation (dpi) versus 5 × 10[7] CFU for the late arrivals; Fig. 3b). We also detected lower densities for the late arrivals than predicted if late arrivals were established in proportion (Fig. 3b, upper right). In the OligoMM[12] mice, the

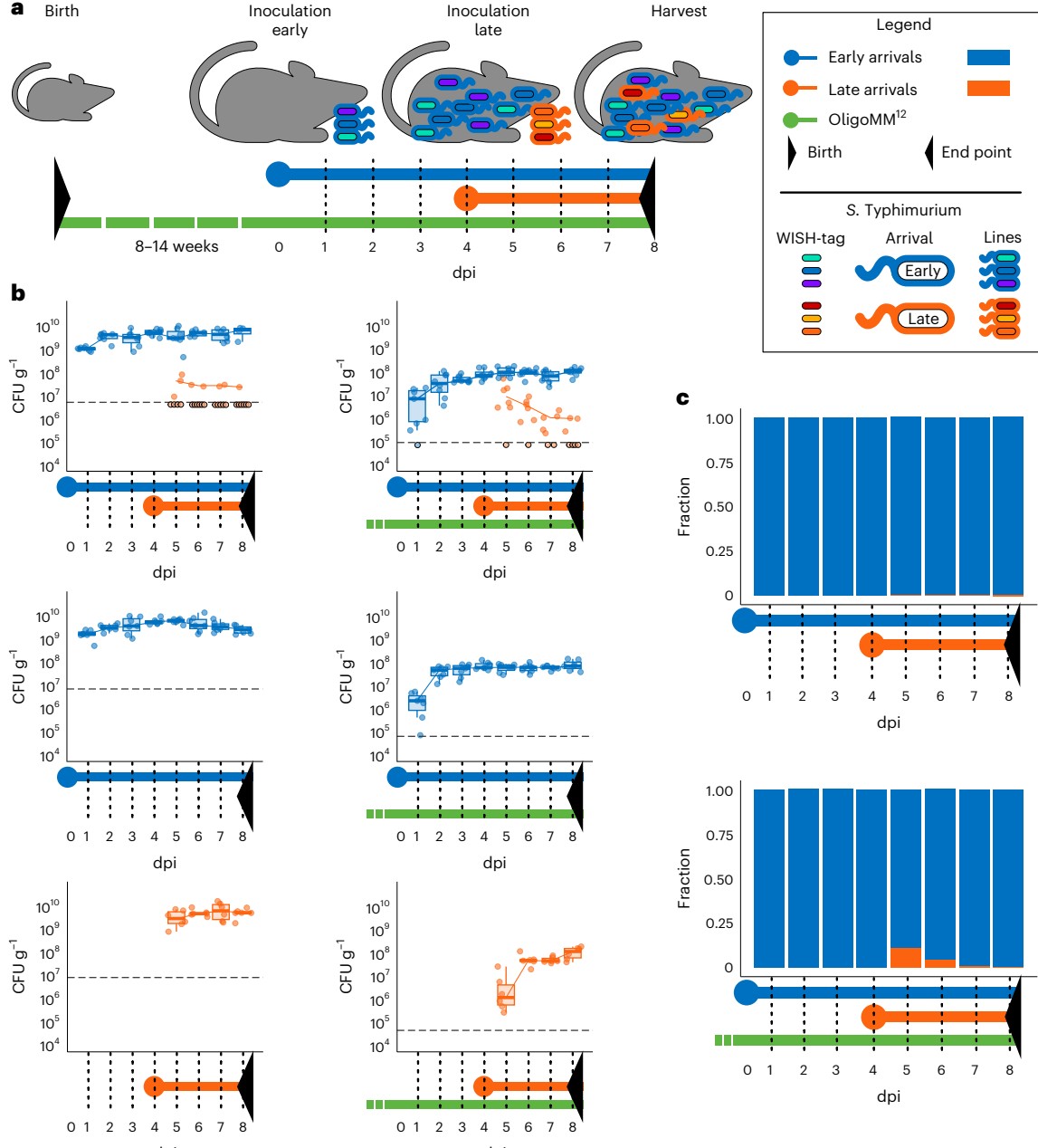

**Fig. 3 | Assessment of the impact of arrival order on intrastrain population dynamics of S. Tm upon early and late arrival in the mouse gut. a**, Scheme illustrating the central treatments. A scheme of all treatments associated with the overall experimental design, including controls, can be found in Extended Data Fig. 2a. Colours represent the different groups of bacteria in the experiment. Points indicate inoculation timepoints, and sampling timepoints are indicated by dotted lines. Black triangles mark the beginning and end of the experiment. The bacterial treatments are illustrated with schematic depictions of S. Tm in the pictures above the bars. **b**, S. Tm density in the collected faeces as determined by plating throughout the experiment. Determination of an association with early or late arrival is based on Illumina sequencing. Each point represents the sum of all reads attributed to WISH-tags belonging to either early or late arrivals in the absence and presence of a synthetic microbiota community, OligoMM[12] (ref. 5). The dashed line represents the detection limit of the WISH-analysis, which is set at 0.0001 of the highest value (corresponding to ~20,000 reads per sample). The centre line of the boxplot represents the median, with the upper and lower borders of the box marking the 25th and 75th percentiles, respectively. Whiskers extend to 1.5 × i.q.r. The number of replicates was $n = 6$ or $n = 7$. **c**, Fractional bar plot indicating the share of early and late arrivals in the population over time. The relative abundance of all WISH-tags for each mouse over time can be seen in Supplementary Fig. 6.

late arrivals were still detected in most mice at day 1 after introduction (Supplementary Fig. 6a). Their proportion of the population decreased each day, although the total S. Tm population remained stable. This indicated that the late arrivals were unable to establish (Fig. 3b,c). Taken together, we found that the late-arriving strains were only transiently present in the gut. Thus, the mouse gut colonization experiments with the avirulent mutant of S. Tm SB300 revealed a pronounced priority effect. The strength of the priority effect and the kinetics of the decline of the late arrivals appear to be modulated by a resident microbiota.

**Dynamics of *Sphingomonas* in the *Arabidopsis* phyllosphere**

As a second and contrasting host, we chose plants. More specifically, we chose the *A. thaliana* phyllosphere to investigate potential intra-strain priority effects. Like the common mouse, this model plant has

been established for microbiome research and a comprehensive strain collection is available[33]. To ensure comparability between the two host model systems, mouse gut and phyllosphere, the experimental design was kept as similar as possible. As the focal strain for plant experiments we chose *Sphingomonas* Leaf257, which has been studied previously for its phenotypic plasticity[52] and is a member of a genus that is ubiquitous in the phyllosphere; that is, part of the core microbiota[53,54]. Consistent with the mouse experiments, we introduced six tags into this focal strain and split the barcoded strains into two sets for early and late arrivals to serve as independent replicates.

We applied two inoculation timepoints, one for the early arrivals and a subsequent one with the late arrivals 7 days later (using 1:1:1 ratios for each, as in the mouse experiments). Because the plants grew considerably between 0 and 7 dpi, and also continuously produced new leaves, the volume of the inoculum was adapted according to the size of the plant (keeping the inoculation population relative to the plant weight constant) (Methods). Continuous sampling from the plant was not possible because sampling for sequencing is destructive. The time series we report here are therefore pseudo-time series from different individuals that were subjected to the same treatment. For technical reasons, we reduced the sampled timepoints to three (Fig. 4a). The isogenic strains were then assessed, both upon colonization of axenic plants and in the presence of a microbiota. For the latter, we used a previously established oligo community composed of 15 members[11].

Experimental procedures and details on mock inoculations and control treatments (24 conditions in total) are given in Methods and Extended Data Fig. 2b.

After inoculation with the early arrivals, the *Sphingomonas* Leaf257 population expanded until the carrying capacity of the plant was reached in less than 7 days (Fig. 4b). After that, the total population expanded and maintained carrying capacity relative to the weight of the growing plant. In mono association, the carrying capacity for *Sphingomonas* Leaf257 was ~$3 \times 10^6$ CFU per plant, although it reached a lower population of ~$7.5 \times 10^5$ CFU per plant when in competition with the oligo 15-species community (Supplementary Fig. 8), as expected[11]. The carrying capacity remained the same as in the wild-type Leaf257 (Supplementary Fig. 9o). In the presence of the synthetic microbiota[11], *Sphingomonas* Leaf257 contributed ~13% to the total population in the 15-species community (Supplementary Fig. 10). The focal strain reached the same carrying capacity, regardless of whether it was inoculated at 0 or 7 dpi, and the mock inoculation at 7 dpi did not reveal any reduction in bacterial populations compared with untreated plants (Supplementary Figs. 9i and 11b,e), indicating that the number of bacteria removed by the second inoculation was below our detection limit.

In the absence of the microbiome, newly introduced members of the community represented ~20% of the total population at 7 dpi, and this proportion remained consistent until 14 dpi (Fig. 4b). When the 15-species community was present on the plants, newly introduced members made up a similar proportion of the total population as the initial members. In both cases, the initial members and the newly introduced members showed an increase in the population. The proportion of WISH-tags stayed the same over the course of the experiment (Fig. 4c and Supplementary Fig. 12). Together, our results revealed striking differences between the two biological systems studied, the gut and the plant; only in the latter did late arrivals establish substantially, whereas they were impaired from doing so in the gut.

## Discussion

There is increasing awareness of the importance of microbiomes, which fulfil essential ecosystem functions. Understanding the rules by which such microbiomes assemble is fundamentally important[55]. In recent years, synthetic communities have emerged for a growing number of environments allowing the generation of comparative data to identify unique and common principles[56]. To enhance the use of these resources, standardized tools that can be applied across systems will facilitate insights not only within, but also among biological study systems.

Here, we describe the development, validation and implementation of a genomic barcoding system that can be used across microbiomes. The concept of genomic barcodes is not new, and they were and are widely used to tackle a plethora of research questions[21,27,28,30,57]. Unlike conventional genomic barcodes, the WISH-tags were specifically designed from the outset for application across microbiomes, with a primary focus on the mouse gut and *A. thaliana* phyllosphere, but with the potential for expansion to other systems by simply adapting the WISH-tag generation pipeline ('Code availability'). By combining the unparalleled dynamic range of qPCR with the cost-effectiveness and scalability of NGS, they open up a multitude of experimental designs without the need to implement two types of barcodes, and readout can be adjusted as needed after preparation of biological samples. To simplify the analysis of barcode-counting in the resulting NGS data, the mBARq tool[58] can be used, enabling user-friendly processing of data output files directly after demultiplexing to obtain counts per barcode and facilitate data analysis.

We thoroughly validated the performance of the WISH-tags at several critical levels, including barcode specificity, high linear performance over a wide range of concentrations and equal amplification. Integrating multiple barcodes in ten different strains from the gut or the phyllosphere confirmed the reliability of WISH-tags in quantifying populations of isogens that differ only in the inserted barcode. This demonstrates that the WISH-tag approach can be used in a wide range of genetically accessible microbial strains from various biological systems.

We analysed priority effects to highlight the versatility of the WISH-tags. Such effects are important to consider in community assembly in any ecosystem because the order of arrival of community members can change the final composition[10,59]; for example, through niche exclusion/pre-emption[60].

Understanding these effects and their underlying causes has important implications. For example, closely related strains can be used to induce colonization resistance against undesirable microbes[61]. In addition, this knowledge can help overcome resistance when introducing a desired species into a resistant community in microbiomes and beyond. This has been particularly explored in plant populations, where the topic has been studied extensively[62]. More specifically, for the mouse gut colonization resistance has been extensively described[63,64], whereas in the phyllosphere several studies investigated bioprotective strains pre-empting or reducing colonization by a pathogen[24,65]. However, the topic remains understudied[24].

Priority effects among different genera or families can be readily investigated by amplicon sequencing because of differences in the 16S/18S ribosomal RNA gene or other markers. Here we investigated intrastrain priority effects of bacteria, which necessitates the tagging of otherwise isogenic strains. Such strains represent the maximal possible niche overlap and a test case to reaffirm a positive correlation between relatedness and the strength of exclusion in the microbiota[66], in line with Darwin's neutralization hypothesis[67] and Freter's niche hypothesis about the co-existence of gut microbiota strains[68]. Here, we selected two focal strains adapted to two different host systems, the animal gut and the phyllosphere, to test within-strain population outcomes. We designed implementation of the experiments to be as similar as possible in terms of the early-arriving and late-arriving populations. In addition, we investigated the effect of the isogens in the absence and presence of a microbiota in both systems.

For the animal gut, we observed rapid clearance of the late arrivals. This clearance was much more pronounced than expected for simple intermixing of early and late arrivals and shows a pronounced priority effect. We noticed that the clearance kinetics of the late arrivals differed between GF and OligoMM[12] mice. In most GF mice, the late arrivals were cleared below the detection limit within 1 day after

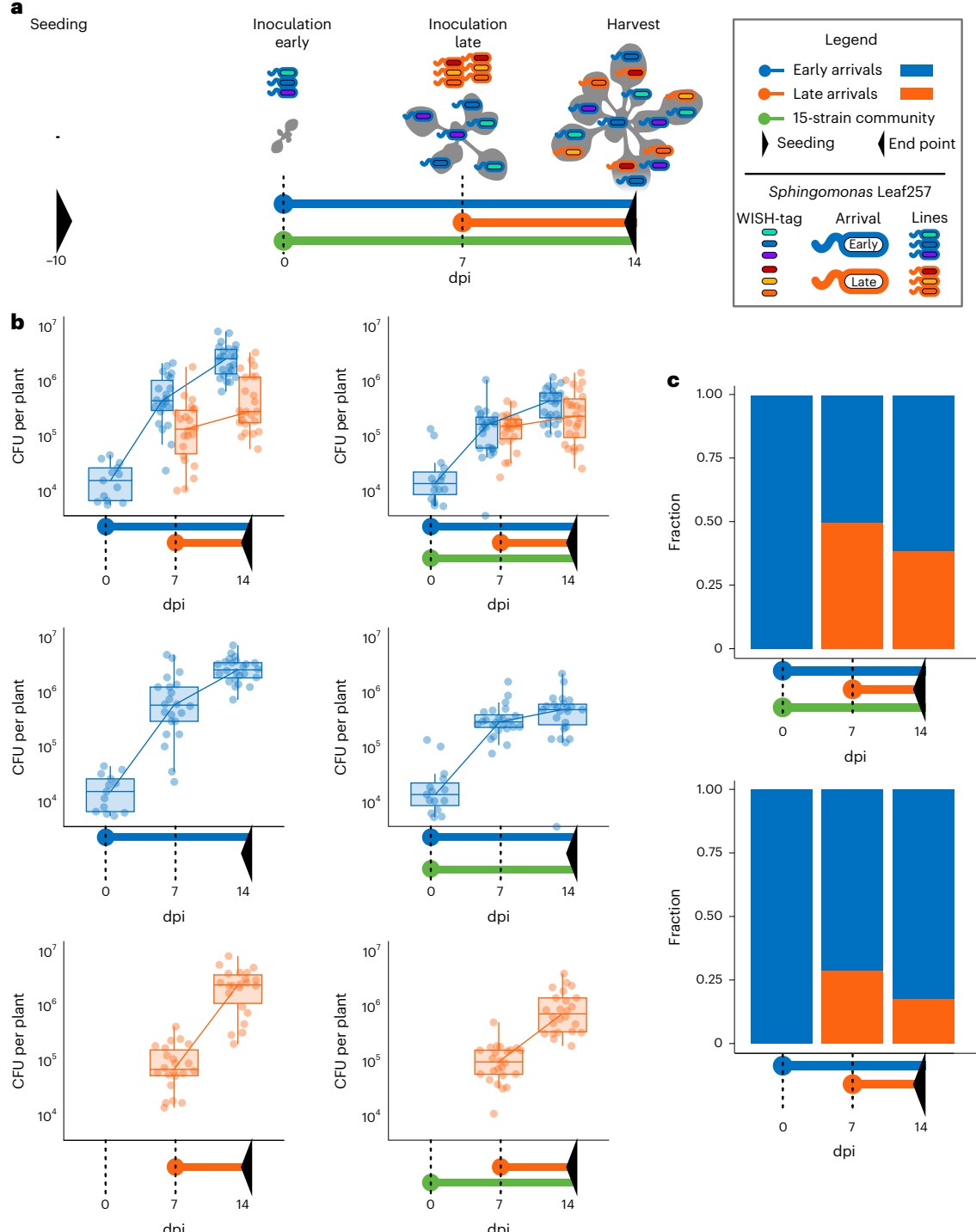

**Fig. 4 | Assessment of the impact of arrival order on intrastrain population dynamics of *Sphingomonas* Leaf257 upon early and late arrival in the phyllosphere. a**, Scheme illustrating the central treatments. A scheme of all treatments associated with the overall experimental design, including controls, can be found in Extended Data Fig. 2b. Colours represent the different groups of bacteria in the experiment. Points indicate when inoculations were performed and sampling timepoints are indicated by dotted lines. Black triangles mark the beginning and end of the experiment. The bacterial treatments are also illustrated with schematic depictions of *Sphingomonas* Leaf257 above the bars. **b**, Pseudo-timelines of Leaf257 populations in the harvested plants as determined by plating. Determination of an association with early or late arrivals is based on Illumina sequencing. Each point represents the reads attributed to WISH-tags belonging to either early or late arrivals in the absence and presence of a synthetic leaf microbiota community, the 15-strain community[11]. The centre line of the boxplot represents the median, with the upper and lower borders of the box marking the 25th and 75th percentiles, respectively. Whiskers extend to 1.5 × i.q.r. The number of replicates was $n = 24$, except for the following: at 0 dpi, $n = 12$; at 7 dpi, $n = 22$ in the top-right and $n = 23$ in the top-left; and at 14 dpi in the bottom-left, $n = 23$. **c**, Fractional bar plot indicating the share of early and late arrivals in the population over time. The relative abundance of all WISH-tags for each plant in this figure can be seen in Supplementary Fig. 12, and comparisons between control treatments can be found in Supplementary Figs. 9 and 11.

their introduction. By contrast, in OligoMM[12] mice, the decrease in late arrivals was less pronounced within 1 day after introduction but became stronger between days 5 and 8. This suggests that the magnitude and kinetics of the priority effect might be modulated by the resident gut microbiota. Our WISH-tagging approach will be a useful tool to investigate the mechanistic underpinnings of this effect of the gut microbiota in future studies.

By contrast, our results from the phyllosphere showed a markedly different result, with the latecomers being able to maintain a substantial portion of the population space. A possible explanation is the opening up of new, uncontested habitats as new leaves emerge and grow, raising interesting questions about bacterial dispersal in the phyllosphere for future studies on spatial aspects and leaf-to-leaf variation.

The striking difference between the outcomes in the two host systems could be caused by their contrasting physical structure, with mixing in the gut facilitating more interactions between competitors in a densely populated habitat, compared with the fragmented habitat on the leaf surface[15]. With our experimental set-up, we cannot rule out a special quality of *Sphingomonas* Leaf257 as the underlying factor, a hypothesis that can be addressed by testing additional strains from the *At*-LSPHERE.

To conclude, we have developed WISH-tags, genomic barcodes allowing their quantification with qPCR and NGS. After insertion into model and non-model bacteria, WISH-tags can be applied to specific microbiomes or across multiple microbiomes. We validated these in several strains in the mouse gut and a plant. Finally, we applied them to address priority effects in commensal strains in the two model systems and found pronounced differences, with the gut microbiome showing a strong colonization resistance compared with the phyllosphere. The use of WISH-tags in future studies may provide valuable insights to uncover compositional rules with substrain-level resolution and advance studies on the underlying mechanisms.

## Methods

### Design and parameters of WISH-tags

The WISH-tag design was based on the requirement to accommodate primers for qPCR and Illumina sequencing (Fig. 1a). The difference between the WISH-tags lies in the 40 bp unique barcode, which is long enough to ensure distinctiveness between barcodes. Inside the barcode region is the binding site for the unique reverse primer, which, together with the universal forward primer, enables qPCR quantification. With the optimal length for qPCR fragments being between 70 and 150 bp (ref. 40), and the most economical/practical length for Illumina sequencing being around 150 bp paired-end reads, we decided on a total size for the WISH-tags of 120 bp, with a qPCR amplicon length of 88 bp. To reduce complexity, the universal forward primer also serves as a primer for Illumina sequencing, the 120 bp amplicon being easily bridged by paired-end reads that can be merged for high accuracy. The necessary spacing is ensured by the inclusion of three spacer regions. To generate the actual sequences, a custom programme was developed that follows the steps outlined in Fig. 1b. The individual parts were all generated separately by creating random strings of bases. These were then filtered to retain only those strings with a GC content between 45% and 55%, and without palindromic sequences. The primers were further required to not align with any of the genomes specified in the exclusion list allowing for up to two errors. Runs of three or more identical bases or identical pairs of bases were also removed, and finally Primer3 (ref. 41) was used to test for even melting temperatures. All parts were then combined to form the complete WISH-tags (Fig. 1a). The sequence motif of the first 500 WISH-tags in Fig. 1c shows the frequencies of the bases along the WISH-tags.

To obtain a number for WISH-tag sequences, the shortest unique part can be used. The 24 bp sequences were generated with the first 19 bp taken from any of the four bases, followed by

3 bp A or T and then 2 bp G or C, making the total number of valid primer sequences:

$$4^{19} \times 2^3 \times 2^2 \approx 8.8 \times 10^{12}$$

The programme log revealed that 13,986 of $1 \times 10^6$ randomly generated primers passed the filtering and alignment steps (~1.4%). Applying 1.4% to the theoretical maximum, it can be estimated that ~$1.23 \times 10^{11}$ primer sequences pass the criteria, for a total of 123 billion possible WISH-tags. The adaptable workflow is available via https://doi.org/10.5281/zenodo.8370066 (ref. 69).

### WISH-tag amplification performance using qPCR

For the validation experiments, plasmids harbouring the individual WISH-tags were ordered from TWIST Biosciences on their high-copy number vector, henceforth referred to as pTWIST. Plasmids pTWIST 1 to 62 were transformed into *E. coli* DH5a. Each strain was validated by Sanger sequencing. To find any irregularities in the amplification profiles of the barcodes, strains harbouring a single pTWIST plasmid carrying one unique WISH-tag were grown individually in 5 ml of lysogeny broth (LB) medium in glass tubes at 37 °C and 200 rpm to start at an optical density at 600 nm ($OD_{600}$) of 0.05 for 16 h. The $OD_{600}$ value of all cultures was measured and normalized to 1 in a volume of 2 ml. Half of the suspension was used for the combinatorial experiment and the remaining 1 ml was used for the signal-to-noise ratio test described below. Overnight cultures of each of the 62 strains were combined so that 62 mixes contained 61 WISH-tags, and 1 mix contained all 62. The plasmids were then purified using the NucleoSpin kit (Macherey Nagel, catalogue no. 740588.50). DNA extracted from a mixture of the entire *At*-LSPHERE collection, *A. thaliana*, the faecal pellets of *M. musculus* C57BL/6 mice carrying a natural and a minimal 12-member community (OligoMM[12]) (ref. 5), was mixed evenly and titrated to a concentration of 10 ng $\mu l^{-1}$. This mix, referred to as BPM-DNA, was aliquoted and used throughout the study. DNA concentration and purity were measured using the Quantus dsDNA kit (Promega). For the qPCR run, three different mixes of WISH-mix and BPM-DNA were run with the following quantities of DNA as a template-mix added to each well: 10 ng $\mu l^{-1}$ BPM-DNA only, 0.02 ng $\mu l^{-1}$ of WISH-mix alone and a mixture of 10 ng $\mu l^{-1}$ BPM-DNA and 0.02 ng $\mu l^{-1}$ WISH-mix. Each of the template-mixes was run separately with one of the specific primers.

For qPCR quantification of the mixes, the QuantStudio 7 Flex Real-Time PCR System (Thermo Fischer Scientific) was used with the FastStart Universal SYBR Green Master (Rox) (Roche) mix. The qPCR programme started with 50 °C for 2 min, followed by 95 °C for 10 min for heat activation of the hot-start polymerase and initial denaturation. The 40 cycles of the PCR ran at a ramp speed of 1.6 °C $s^{-1}$, starting with denaturation at 95 °C for 15 s, then elongation 60 °C for 60 s, and finally 95 °C for 15 s. Each qPCT run was followed by a melt curve to assess the quality of the run. The settings for the melt curve are 60 °C for 60 s with a slow increase, at 0.05 °C $s^{-1}$ ramp speed, to 95 °C. The primers employed for quantification of WISH-tag abundance were the Universal_fwd primer in conjunction with the specific reverse primer for the WISH-tag targeted. Two technical replicates of each reaction were run, 2 µl of template were used.

### WISH-tag amplification performance using NGS

*E. coli* DH5α strains harbouring the pTWIST plasmids 1 to 62 were used (Supplementary Table 1). *E. coli* cultures were grown overnight, after which the $OD_{600}$ value for 62 *E. coli* DH5α carrying individual WISH-tags on their pTWIST plasmids were normalized to an $OD_{600}$ of 1. They were then combined in equal volumes into sets of eight (Extended Data Fig. 1a,b), henceforth called a mix. We used the Macherey Nagel NucleoSpin Kit to extract the plasmid DNA from the mixes, and then evaluated DNA quality using the NanoDrop ND-1000 (Thermo Fisher Scientific). Each mix was prepared separately for Illumina sequencing and its own set of indices.

For evaluation of the linearity of the NGS readout across dilutions, mixes were combined according to Extended Data Fig. 1 across eight libraries (A to H). The DNA of the mixes was then extracted using the NucleoSpin kit (Macherey Nagel, catalogue no. 740588.50), after which we added the BPM-DNA at 10 ng μl$^{-1}$. Subsequently, we individually prepared sequencing libraries from the eight libraries. Short-read sequencing was performed by Novogene UK and run on a NovaSeq 6000 from which 97.3 Gb of data or ~5 million reads per mix were generated, the vast majority of which had a Phred score above 30. Subsequently, the reads were demultiplexed and grouped by WISH-tags using mBARq software[58], where misreads or mutations of up to five bases were assigned to the closest correct WISH-tag sequence. These results were normalized to the total number of sequences per library, and used to create Fig. 2b–d.

## Generation of WISH-tagged strains

*S.* Tm ATCC14028S and SL1344, as well as *E. coli* 8178, were grown in LB media or on LB agar plates, containing the appropriate antibiotics (15 μg ml$^{-1}$ chloramphenicol (ROTH); 50 μg ml$^{-1}$ streptomycin (ROTH); 100 μg ml$^{-1}$ ampicillin (ROTH)) at 37 °C (strains containing pSIM5 were grown at 30 °C).

WISH-tags were amplified from pTWIST using high-fidelity Phusion polymerase (Thermo Fisher Scientific) and integrated into *S.* Tm SL1344 (strain SB300) and ATCC14028S, as well as *E. coli* 8187 using the λ-red system[43] and pSIM5 (ref. 44). Integration was targeted to a fitness neutral locus between the pseudogenes *malX* and *malY* as described previously[21]. Correct integration was checked via colony PCR and WISH-tags were verified by sequencing (Microsynth AG). The avirulent target strain M2702 (SL1344 Δ*invG* Δ*ssaV*) was tagged via P22-phage transduction. For each WISH-tag, two clones of independent transductions were stocked and checked for correct integration via colony PCR.

The actual WISH-cassette on the pTWIST plasmids was flanked by two BsaI recognition sites with the restriction sites lying inward, which were used to create sticky ends complementary to EcoRI and BamHI (Supplementary Fig. 13). The integration vector p7XX00 was a suicide plasmid with the R6K origin of replication that depends on the *pir* gene for replication, turning it into a suicide vector for almost all target strains[69]. It was originally called pUC18R6KT-mini-Tn7T[48], and is available with kanamycin, streptomycin, gentamycin[70,71] and tetracycline[72] resistances for selection of successful integration into varied target strains. In addition, the backbone of the vector harbours an ampicillin resistance. The integration cassette contained one EcoRI and one BamHI restriction site, which were used to integrate the WISH-cassette from the pTWIST. All enzymes used were procured from New England Biolabs and used in accordance with the manufacturer's instructions.

Integration was not possible for *Rhizobium* Leaf68 Tn7, therefore flanks homologous to a silent region of the genome were used to replace the Tn7 flanks for integration by homologous recombination. The 500 bp flanks were amplified from the target species genome using primers that added overhangs including BsaI restriction sites. These were then used to generate sticky ends compatible with EcoRI and BamHI. In a four-part one-pot assembly, the vector, cut as described previously, the two flanks and the WISH-cassette were ligated to create the integration vector.

An EcoRI/BamHI double-digest was used to create the linearized vector of the respective p7XX plasmid to ligate the WISH-cassette, cut from pTWIST by BsaI. For the ligation, the two digest solutions were combined in one tube and NEB T4 ligase and ATP were added. Five microlitres of the ligation were then transformed into *E. coli* BW23474, a strain for high-copy expression of *pir*-dependent suicide plasmids, and plated on selective LB agar plates. Following the appropriate incubation times, single colonies were picked and grown overnight in LB liquid media under selection. The plasmids were then extracted using the NucleoSpin kit (Macherey Nagel, catalogue no. 740588.50) and verified by Sanger sequencing (Microsynth AG).

For the species *Duganella* Leaf61, *Methylobacterium* Leaf88, *Xanthomonas* Leaf131, *Xylophius* Leaf220 and *Sphingomonas* Leaf257, Tn7-mediated integration next to the *glmS* gene in the genome was performed. The validation primers associated with each species (Supplementary Table 1) were used to verify correct integration by running the PCR products on agarose gels and comparing the resulting bands with the wild-type band and the expected fragment size. *Methylobacterium* Leaf88 possesses two homologues of the *glmS* gene in which the Tn7 transposase facilitates integration at different frequencies. Clones of labelled Leaf88 were screened until each isogenic line was integrated once into the most frequently occurring integration site. For *Rhizobium* Leaf68 homologous recombination was employed for the insertion. Because of the lack of a usable Tn7 site, a new site had to be chosen. To limit the potential fitness effects, an intergenic region between two genes behind their terminators was chosen. Several potential sites were tested. The site between the putative protein coding sequences ASF03_RS09010 and ASF03_RS09015 was fitness neutral (Supplementary Fig. 5).

## Mouse experiments

For the animal experiments, 8–14-week-old male and female mice were assigned randomly to experimental groups with no bias for sex. All mice originate from C57BL/6 originally obtained from Jackson Laboratories. Mice were kept under specific pathogen-free conditions in individually ventilated cages in the Rodent Center HCI of the ETH Phenomics Center (EPIC) technology platform of ETH Zurich (12:12 h light/dark cycle, 21 ± 1 °C) during the course of the experiment. GF C57BL/6 mice were bred in flexible film isolators at the EPIC isolator facility at ETH Zurich. OligoMM[12] mice are ex-GF C57BL/6 mice, which were colonized with a defined set of 12 bacterial strains isolated from the murine gut[5]. They were bred in flexible film isolators at the EPIC isolator facility. All animal experiments, including those validating *E. coli* 8178, *S.* Tm SL1344 and ATCC14028S, underwent review and approval by the Tierversuchskommission and Kantonales Veterinäramt Zürich. These experiments were conducted under licences ZH158/2019, ZH108/2022 and ZH109/2022, and followed both cantonal and Swiss legislation.

## Evaluation of strain fitness for mouse gut strains

To evaluate the fitness of the barcoded mouse gut strains, they were tested in seven OligoMM[12] mice. The near-isogenic strains all exhibited the same relative fitness, aside from the one line of wild-type *S.* Tm barcoded with WISH-207, which behaved as the other lines at first but was always reduced on day three and unrecoverable after that. Because ATCC14028S is a pathogenic strain that causes inflammation on day two to three, we assume that this strain is afflicted by an off-target effect that reduces its fitness in the inflamed gut. *S.* Tm SL1344 was tagged with the same nine unique barcodes and tested in three different mouse backgrounds, 129SvTvTac, C57BL/6 and OligoMM[12], and all of them showed the same relative fitness over the duration of the experiments. The mouse commensal *E. coli* 8178 was used to create four lines carrying different barcodes, which were tested in C57BL/6. The endpoint measurement revealed that all lines behaved the same. The avirulent *S.* Tm SL1344 Δ*invG* Δ*ssaV* was not tested separately, but as a control in the experiment described below, confirming that no fitness cost is imposed by the tag (Supplementary Fig. 3).

## Plant experiments

*Arabidopsis thaliana* Col0 was used for all plant experiments. Plants were grown in six-well plates (Techno Plastic Products), filled with 5 ml of heat-sterilized, calcined clay (Cremonini terre rosse) and 2.5 ml of sterile filtered ½ Murashige–Skoog medium containing vitamins (Ducefa, catalogue no. M0222)[73]. Percival CU41-L4 growth chambers were used and were set to 23 °C and 54% relative humidity, with 11 h of light (220 μmol m$^{-2}$ s$^{-1}$ of visible light (Sylvania Reptistar F18W/6500K) and 4.2 μmol m$^{-2}$ s$^{-1}$ of UV light (Philips Master TL-D

18 W/950 Graphica)) and 13 h of darkness. A single, sterilized seed was placed in the centre of each well and grown for 7 days. At this point, plants from extra plates were transferred to wells with ungerminated seeds (typically <5%).

For evaluation of the relative fitness of the barcoded *Sphingomonas* Leaf257 lines, plants were inoculated with bacteria, 10 days after sowing the seeds. The strains used in this experiment were streaked individually on R2A (Merk, catalogue no. 17209) plates with 0.5% methanol and grown for 2 days. The strains were then resuspended individually in 10 mM $MgCl_2$. The $OD_{600}$ was normalized to 0.5 for each strain. Strains were then mixed according to the experiments and finally diluted to $OD_{600}$ of 0.02 for inoculation. Each plant was inoculated using 50 µl of the suspension, which was distributed evenly across the entire plant via droplets placed by pipette.

### Evaluation of relative fitness of isogenic lines for *At*-LSPHERE strains

Six WISH-tags were integrated for each of the species from the *At*-LSPHERE collection. Distinct barcodes were used for each species so that information on the species identity can be retrieved in future experiments, if they were mixed.

All isogens of the six species were tested for fitness defects after integration using in vitro growth experiments. For these, the isogenic lines and the wild-type for each species were inoculated in 40 ml of liquid R2A in 250 ml baffled flasks and grown until the stationary phase as precultures. The barcoded lines of one species were then combined at even ratios based on assessment of the $OD_{600}$. To gain information about the relative fitness of the barcoded lines compared with the wild-type, this mixture was then combined with a culture of the wild-type at ratios of 100%, 10%, 1%, 0.1% and 0.01% barcoded. For *Xanthomonas* Leaf131, only 10%, 0.01% and 0.001% were included. The strains were mixed at the described ratios and inoculated into 40 ml of R2A liquid media in 250 ml baffled flasks. Bacterial cultures were grown at 22 °C and 180 rpm. After reaching a high $OD_{600}$ value (depending on the species), samples were collected and a dilution series was spotted on selective and non-selective R2A square plates to determine CFU and the wild-type to barcoded strain ratio. All barcoded strains used here harbour a selection cassette, making determination of their fraction in the overall population straightforward. An additional volume of the culture was used to extract the DNA of the strains for quantification (FastDNA spin kit for soil, MP Bio). Subsequently, qPCR was used to quantify the WISH-tags.

*Sphingomonas* Leaf257 isogens were also tested in an in planta validation experiment. All steps were performed as described in the section 'Plant experiments'. The six lines were mixed at an even ratio and subsequently used to inoculate axenic *A. thaliana* plants. After 2 weeks of growth, the plants were harvested, and used for short-read sequencing to quantify the barcoded lines from the treatment with a fully labelled population (Supplementary Fig. 4). All could be recovered at relative abundances similar to those they had in the inoculum, confirming fitness neutrality.

### Sequential arrival of bacterial populations in mouse experiments

In preparation for the mouse experiments, *S.* Tm or *E. coli* cultures were grown overnight in LB medium supplemented with the appropriate antibiotics at 37 °C with shaking, 1:100 diluted in fresh LB medium and grown for 4 h at 37 °C with shaking. The cells were centrifuged and washed twice in phosphate-buffered saline (PBS; 2.7 mM KCl, 1.8 mM $KH_2PO_4$, 137 mM NaCl, 10 mM $Na_2HPO_4$). The mice were infected with ~5 × $10^7$ CFU *S.* Tm via oral gavage in 100 µl aliquots. Mock-treated mice were gavaged with 100 µl of PBS. Faecal samples were collected every 24 h. Mice were euthanized 8 days after infection by $CO_2$ asphyxiation. Caecal content (CC) as well as liver, spleen and mesenteric lymph node samples were collected. Faecal, liver and spleen samples were homogenized by steel ball beating in 1,000 µl of PBS, and CC and mesenteric

lymph node in 500 µl of PBS, using a tissue-lyser (Qiagen). Bacterial loads were determined by plating on MacConkey agar plates containing the appropriate antibiotics. Mouse AW229 was removed from the analysis because of clearing of all *S.* Tm from the gut for unknown reasons. To evaluate inflammation levels in the mice, faecal lipocalin-2 levels were measured in samples homogenized in 1,000 µl of PBS using the mouse lipocalin-2/NGAL DuoSet ELISA kit (R&D Systems, catalogue no. DY1857) (Supplementary Fig. 7). In total, 21 OligoMM[12] and 21 GF mice were used in the experiments. Eight mice in the GF Early-Late, one of which (AW229) was removed from the downstream analysis because of clearing of *S.* Tm from the gut. There were six mice in the GF PBS-Late group, whereas the GF Early-PBS group contained seven mice. Eight mice were used in the Oligo Early-Late group. The Oligo PBS-Late and Oligo Early-PBS groups contained six and seven mice, respectively.

### Sequential arrival of bacterial populations in plant experiments

Preparation of the strains for the in planta experiments to assess intra-strain dynamics followed the same steps as described above. After restreaking the WISH-tagged Leaf257 isogenic lines from their individual glycerol stocks on R2A + M plates, they were grown at 22 °C for 2 days. The strains were then resuspended in 10 mM $MgCl_2$ and the $OD_{600}$ normalized to 0.5. The three strains belonging to Set1 (WISH_64, WISH_65, WISH_66) and Set2 (WISH_67, WISH_68, WISH_69) were then mixed in equal volumes. They were then mixed again in equal volumes with sterile 40% glycerol solution which was aliquoted and stored at −80 °C to ensure that the ratios of the WISH-tags between the two experimental runs remained the same. Plants were grown axenically for 10 days in the first replicate and, because they grew more slowly, for 11 days in the second replicate. They were then inoculated with the WISH-strains for the early arrivals, Set1 for the first run and Set2 for the second run, as well as the community strains, where applicable. All 24 different treatments are listed in Extended Data Fig. 2b. The day of the initial inoculation is referred to as 0 dpi. Because of the slower growth in the second experimental run, the first inoculation was shifted from day 14 after seeding to day 15. The impact of the slower growth is still significant (Supplementary Fig. 14) but does not impact our results (Supplementary Fig. 11). Regarding the ratio of inoculum to plant weight, the plants were 2.9 times larger at 7 dpi, whereas 5 times more bacteria were used for the inoculation. This resulted in 8.7 times the number of bacteria sticking to the plants, so 3 times more per plant weight (Supplementary Figs. 9 and 11).

The Leaf257 WISH isogenic line inoculum was prepared from the pre-made, frozen mixes for Set1 or Set2, which were thawed on ice, washed by centrifugation (2 min at 5,000*g*) and subsequently resuspended in fresh 10 mM $MgCl_2$. The $OD_{600}$ was adjusted to 0.5 by dilution in 10 mM $MgCl_2$.

The inoculant for the 15-species community was freshly prepared by restreaking the individual species from glycerol stock and growing them on R2A + M plates for 2–5 days depending on the species. On the day of inoculation, the species were resuspended in $MgCl_2$ and the $OD_{600}$ of all inoculates was subsequently set to 0.5. In the inoculum for conditions containing the 15-species community, as well as for the WISH isogenic lines of Leaf257, each of the two suspensions individually accounted for an $OD_{600}$ of 0.05 for a total $OD_{600}$ of 0.1. In all conditions where only Leaf257 was added, the $OD_{600}$ in the total inoculum was 0.05. The inocula were then distributed over the plants according to Extended Data Fig. 2b. The concentration of bacteria was the same for both inoculation time-points, but 4 µl was used 0 dpi and 20 µl at 7 dpi to achieve a similar ratio of inoculum to plant weight. Approximately 1 h after inoculation, conditions 9, 18 and 23 were harvested (Extended Data Fig. 2b). The plants were then grown for an additional 7 days until the second inoculation, after which several additional conditions were harvested (Extended Data Fig. 2b). The preparations were performed as described before, and the plants were inoculated as shown in Extended Data Fig. 2b. Plants were grown a further 7 days until the time of the final harvest.

We planned 12 replicates per treatment per repetition of the experiment, which should result in 24 independent plants for each treatment. Some plants were not harvested because of a strong deviation from the normal phenotype, meaning they were either very small or had deformed leaves. For each treatment, ≥20 plants were analysed for colonization.

### Harvest protocol
Before the harvest, 2 ml Eppendorf tubes were prefilled with 600 µl of 100 mM phosphate buffer (pH7), labelled and their weights recorded (Mettler Toledo, XS205 Dual range). The phyllospheres were harvested by cutting the stem below the cotyledons with a scalpel, using tweezers to bend the leaves back for better access. Particles of calcinated clay sticking to the plants were removed and the plants were transferred to their respective tubes. The tubes were weighed again and, after ensuring complete submersion of the plants, were sonicated for 7 min at 47 kHz (Branson 2210). Tubes with the buffer containing washed-off bacteria were vortexed, and 500 µl was immediately transferred to a new, prelabelled 1.5 ml Eppendorf tube, which was snap-frozen in liquid nitrogen. The remaining ~100 µl was transferred to a 96-well plate, in which a tenfold dilution series was performed, using the same phosphate buffer as for the sonication. CFU assessment was performed by spotting 4 µl of each dilution step on R2A + M plates. For selective spotting, the appropriate antibiotic was used for each strain (Supplementary Table 1). After a suitable incubation period, the colonies were counted and the CFU per gram of plant fresh weight was calculated.

### DNA extraction mouse faeces
Enrichment cultures of faecal samples were set up by inoculation of 5 ml of LB liquid cultures containing appropriate antibiotics with 100 µl of homogenized faecal samples. Cultures were grown overnight at 37 °C with shaking and supernatant was removed via centrifugation for 3 min at maximum speed using a table-top centrifuge. Genomic DNA was extracted using the QIAmp DNA Mini Kit (Qiagen).

### DNA extraction plant wash-off
Frozen phyllosphere wash-off samples were thawed on ice. DNA was extracted as described in the MasterPure Complete DNA and RNA Purification Kit (Lucigen, catalogue no. MC85200) using the protocol for cell samples. To improve the efficiency of the extraction protocol, which requires a minimal amount of DNA, 400 µl of $OD_{600} = 1$ Leaf257 overnight culture (untagged) in liquid R2A + M was systematically added to each sample before extraction. After extraction, the samples were transferred into 200 µl PCR tubes with individual lids for storage and easy access.

### Library preparation
Libraries for short-read NGS sequencing were prepared using a two-stage, PCR-based approach. The first stage of PCR was performed using universal forward and reverse primer (Supplementary Table 1) with 25 cycles of: initial denaturation 94 °C for 2 min, denaturation 94 °C for 30 s, annealing 55 °C for 30 s, extension at 72 °C for 60 s and a final extension at 72 °C for 10 min followed by an infinite hold at 4 °C. Primers were then removed from each sample using Exonuclease I and Antarctic Phosphatase (NEB, catalogue no. M0289S). Conditions for the digest were 30 min at 37 °C followed by enzyme deactivation at 80 °C for 15 min. The barcoding step of the PCR used the extended Unique dual indexing primer set from short-read NGS. After initial denaturation at 94 °C for 2 min, PCR conditions were identical to the first PCR reaction but ran for only ten cycles. After each PCR step, 5 µl of each sample was loaded onto a 2% agarose gel with 0.01% GelRed (Merck, catalogue no. SCT123) for quality control. A second gel was also used to assess the PCR product concentration to allow the addition of approximately equal proportions of each sample amplicon for pooling. According to the intensity of the product band in the second gel, 192 samples were pooled so that each sample had approximately the same concentration for a total of three libraries. In case little or no amplification product was visible (for example, for controls), all the available volume of samples was added to the libraries. Magnetic beads (Sera-Mag SpeedBeads carboxylate modified; Thermo Fischer Scientific) were used to remove primers and clean the libraries. The quality of all three libraries was then verified using a 4200 TapeStation and D5000 ScreenTape (Agilent). For the validation experiments, libraries were sequenced by Novogene UK Ltd to achieve 0.5 GB output on the NGS NovaSeq platform, 150 bp paired-end sequencing. Samples from the priority effect experiments were sequenced at the Genetic Diversity Center ETH Zurich (GDC) using Illumina MiSeq and the v2 reagent kit.

### Primers and oligonucleotides
All primers and oligonucleotides used in this study are listed in Supplementary Table 1 under the heading 'Primers'.

### Imputation of relative abundance to CFU
To illustrate the relative CFU for the early-arriving and late-arriving strains in the in vivo experiments in mouse and plant in Figs. 3a and 4a, we calculated the fraction of the total population composed of early and late arrivals and applied these factors to the CFU data received directly from the plant. The relative abundance of the early and late arrivals can be seen in Figs. 3c and 4c, corresponding to the data shown in Figs. 3b and 4b.

### Statistics and reproducibility
No statistical method was used to predetermine the sample size. Instead, we made an informed decision based on typical sample sizes used in previous studies employing comparable experimental set-ups for *Arabidopsis*[10–12], and for experiments in mice[21,34,35]. One mouse was excluded from the analysis because of the unexplained clearance of *S*. Tm from the gut. For the plant experiments, samples were excluded based on the physical appearance of the plants, meaning they looked diseased, dead or their development was severely delayed. The experiments were not randomized. For the plant experiments, the experimenter was fully blind to the allocation of treatments after inoculation. Because of the nature of the mouse experiments, the experimenter could not be blinded to the treatment allocation. For statistical analysis of the resulting data, we used R statistical software. The exact code and packages used can be found on Zenodo under https://doi.org/10.5281/zenodo.10489293 (ref. 74). Various statistical models were first assessed using the Akaike information criterion to analyse the data. The most appropriate models were then selected for comparison. Equal variance was confirmed using the Bartlett test, and log-normal distribution was confirmed using the Shapiro test. After meeting these criteria, the Welch-pairwise test was employed, and the resulting *P* values were corrected using the Bonferroni method.

### Reporting summary
Further information on research design is available in the Nature Portfolio Reporting Summary linked to this article.

## Data availability
Raw reads of amplicon samples can be found in the European Nucleotide Archive (ENA) under accession number PRJEB66333. All other raw data are available on Zenodo under https://doi.org/10.5281/zenodo.10489293 (ref. 74). Strains used in this study are available upon request. Source data are provided with this paper.

## Code availability
The code used for the generation of the WISH-tags is available on Zenodo under https://doi.org/10.5281/zenodo.8370066 (ref. 69). The code used to analyse all data and generate figures can be accessed at https://doi.org/10.5281/zenodo.10489293 (ref. 74). No unpublished algorithms or methods were used.

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

## Acknowledgements

We thank M. Bortfeld-Miller, B. Rast and C. Lefebvre for their assistance in plant harvesting of plant experiments. We also appreciate the insightful feedback on design features, as well as test samples, provided by M. Arnoldini. We acknowledge the Genetic Diversity Center (GDC), ETH Zurich, for their collaboration in generating sequencing data, with special thanks to S. Kobel for her assistance. This work was supported by the National Centre of Competence in Research Microbiomes, funded by the Swiss National Science Foundation (51NF40_180575) to S.S., W.-D.H. and J.A.V.

## Author contributions

B.B.J.D. was responsible for the study concept, WISH-tag design, strain tagging (*At*-SPHERE strains), experimental design, plant experiments, analysis, data visualization and writing the paper. Y.S. was responsible for the experimental design, strain tagging *S*. Tm, mouse experiments and reviewing the paper. B.D.N. was responsible for strain tagging *S*. Tm and the experimental design. A.S. undertook data analysis, visualization, software development and reviewing the paper. C.M.F. was responsible for the WISH-tag design and analysis. C.S. undertook strain tagging and validation of *S*. Tm SL1344 and ATCC14028S. Y.C. undertook strain tagging and validation of *E. coli* 8178. S.S. and W.-D.H. were responsible for the study concept, experimental design, supervision and reviewing the paper. J.A.V. was responsible for the study concept, experimental design, supervision and writing the paper. All authors read the paper and approved its submission.

## Funding

## Competing interests

The authors declare no competing interests.

## Additional information

**Extended data** is available for this paper at https://doi.org/10.1038/s41564-024-01634-9.

**Correspondence and requests for materials** should be addressed to Julia A. Vorholt.

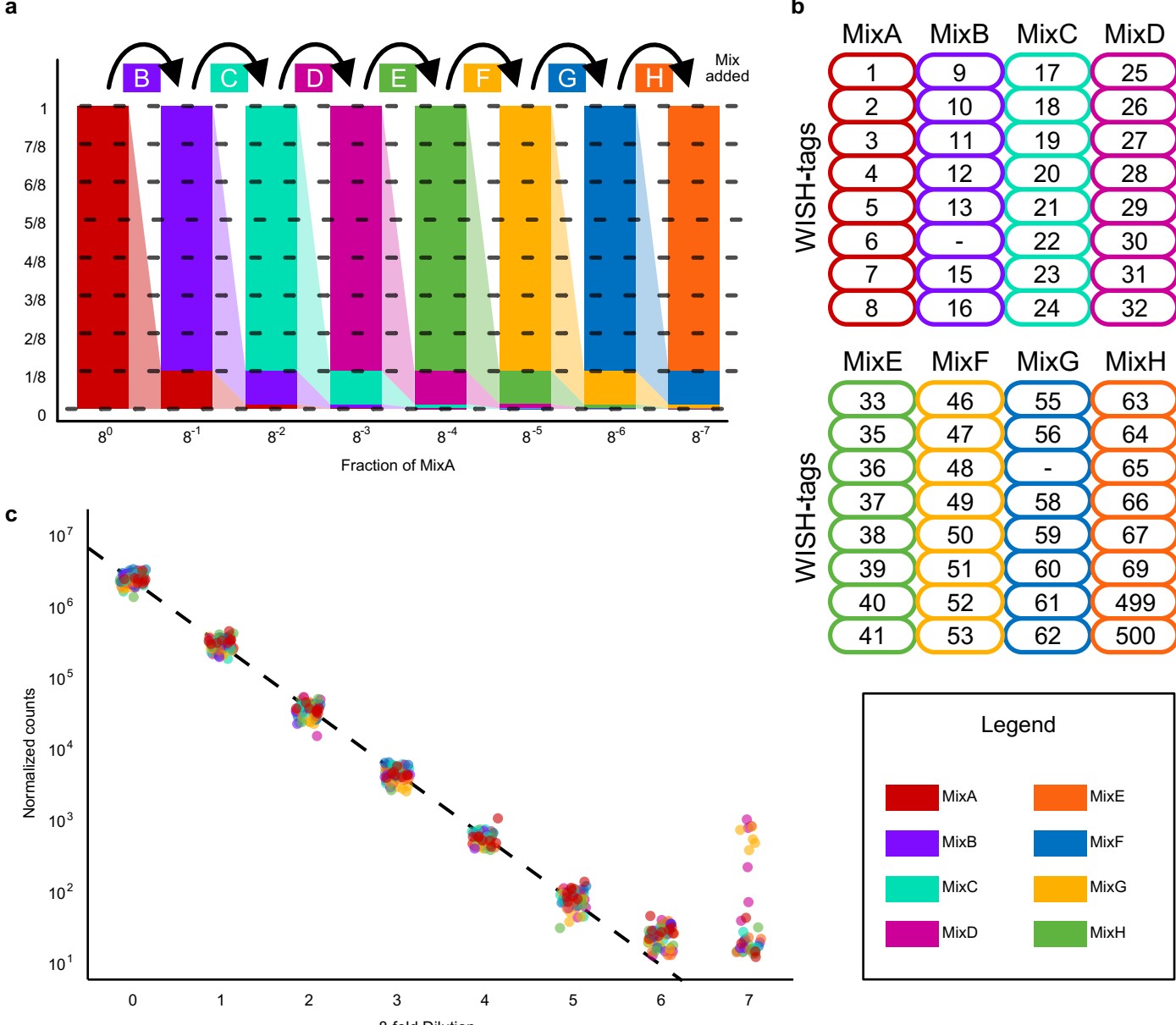

**Extended Data Fig. 1 | Setup and additional information on the Illumina sequencing-based validation experiments for the WISH-tags. (a)** Illustration of the dilution process for the deep sequencing experiment to assess linear amplification over 7 orders of magnitude. The arrows on top indicate which mix of WISH-tags was added progressively from the left to the right. The fractions on the x-axis indicate the fraction of MixA in the total mix used for library preparation, which was generated from dilution 8 of each dilution series, shown here exemplarily for Library H. **(b)** List of the WISH-tags that were included in each mix, with WISH-tags 14 (MixB) and 57 (MixG), missing due to slow growth of the carrier-strains in the pre-culture. **(c)**. Jitter plot showing the counts for each tag. Jittering on the x-axis was introduced to improve the visibility of the dots. The dotted line indicates the theoretical number of reads expected at each dilution (n = 62 for each dilution, as well as each mix).

## a Mouse

| Condition | Birth | 0 dpi | 4 dpi | 8 dpi |
|---|---|---|---|---|
| 1 | GF | Early | Late | endpoint |
| 2 | GF | Early | Mock | endpoint |
| 3 | GF | Mock | Late | endpoint |
| 4 | OligoMM12 | Early | Late | endpoint |
| 5 | OligoMM12 | Early | Mock | endpoint |
| 6 | OligoMM12 | Mock | Late | endpoint |

## b Plant

| Condition | 0 dpi | 7 dpi | 14 dpi |
|---|---|---|---|
| 1 | Early | Late | Harvest |
| 2 | Early | Mock | Harvest |
| 3 | Early | None | Harvest |
| 4 | Mock | Late | Harvest |
| 5 | Mock | Late — Harvest | |
| 6 | Early | Late — Harvest | |
| 7 | Early | Mock — Harvest | |
| 8 | Early | None — Harvest | |
| 9 | Early — Harvest | | |
| 10 | Early / Com | Late | Harvest |
| 11 | Early / Com | Mock | Harvest |
| 12 | Early / Com | None | Harvest |
| 13 | Com | Late | Harvest |
| 14 | Com | Late — Harvest | |
| 15 | Early / Com | Late — Harvest | |
| 16 | Early / Com | Mock — Harvest | |
| 17 | Early / Com | None — Harvest | |
| 18 | Early / Com — Harvest | | |
| 19 | Mock | Mock | Harvest |
| 20 | None | None | Harvest |
| 21 | Com | None | Harvest |
| 22 | Com | Harvest | |
| 23 | Com — Harvest | | |
| 24 | WT 257 | None | Harvest |

**Extended Data Fig. 2 | Overview of experimental design in the experiments for the evaluation of the impact of arrival order on the in-situ population dynamics.** (**a**) List of all treatments (1-6) of the mouse gut priority effect experiment. All treatments were repeated, inversing the WISH-tags constituting early and late arrivals of tagged *S*. Tm strains (labelled 'Early' and 'Late'). (**b**) List of all treatments (1-24) of the in-planta experiment. Synthetic communities are indicated in green (OligoMM12 for mouse (ref. 5) and 'Com' for 15 strain community (ref. 11)). GF, germ-free.

# Reporting Summary

## Statistics

For all statistical analyses, confirm that the following items are present in the figure legend, table legend, main text, or Methods section.

| n/a | Confirmed | |
|---|---|---|
| ☐ | ☒ | The exact sample size (*n*) for each experimental group/condition, given as a discrete number and unit of measurement |
| ☐ | ☒ | A statement on whether measurements were taken from distinct samples or whether the same sample was measured repeatedly |
| ☐ | ☒ | The statistical test(s) used AND whether they are one- or two-sided<br>*Only common tests should be described solely by name; describe more complex techniques in the Methods section.* |
| ☐ | ☒ | A description of all covariates tested |
| ☐ | ☒ | A description of any assumptions or corrections, such as tests of normality and adjustment for multiple comparisons |
| ☐ | ☒ | A full description of the statistical parameters including central tendency (e.g. means) or other basic estimates (e.g. regression coefficient) AND variation (e.g. standard deviation) or associated estimates of uncertainty (e.g. confidence intervals) |
| ☐ | ☒ | For null hypothesis testing, the test statistic (e.g. *F*, *t*, *r*) with confidence intervals, effect sizes, degrees of freedom and *P* value noted<br>*Give P values as exact values whenever suitable.* |
| ☒ | ☐ | For Bayesian analysis, information on the choice of priors and Markov chain Monte Carlo settings |
| ☒ | ☐ | For hierarchical and complex designs, identification of the appropriate level for tests and full reporting of outcomes |
| ☒ | ☐ | Estimates of effect sizes (e.g. Cohen's *d*, Pearson's *r*), indicating how they were calculated |

*Our web collection on statistics for biologists contains articles on many of the points above.*

## Software and code

Policy information about availability of computer code

| Data collection | Quant Studio Real Time PCR1 version 1.3 |
|---|---|
| Data analysis | R Studio running R version 4.2.1 (all code written and used for the analysis of data generated from this study can be found on Zenodo under the following DOI: 10.5281/zenodo.8370066), Excel 365, mBARq version 1.0.0 (https://github.com/MicrobiologyETHZ/mbarq) |

For manuscripts utilizing custom algorithms or software that are central to the research but not yet described in published literature, software must be made available to editors and reviewers. We strongly encourage code deposition in a community repository (e.g. GitHub). See the Nature Portfolio guidelines for submitting code & software for further information.

## Data

Policy information about availability of data

All manuscripts must include a data availability statement. This statement should provide the following information, where applicable:
- Accession codes, unique identifiers, or web links for publicly available datasets
- A description of any restrictions on data availability
- For clinical datasets or third party data, please ensure that the statement adheres to our policy

Raw reads of amplicon samples can be found in the European Nucleotide Archive (ENA) under accession number PRJEB66333. All other raw data together with the code necessary to generate any figures in this publication from it are available on Zenodo under the following DOI: 10.5281/zenodo.8370066

# Research involving human participants, their data, or biological material

Policy information about studies with [human participants or human data](). See also policy information about [sex, gender (identity/presentation), and sexual orientation]() and [race, ethnicity and racism]().

| | |
|---|---|
| Reporting on sex and gender | N/A |
| Reporting on race, ethnicity, or other socially relevant groupings | N/A |
| Population characteristics | N/A |
| Recruitment | N/A |
| Ethics oversight | N/A |

Note that full information on the approval of the study protocol must also be provided in the manuscript.

# Field-specific reporting

Please select the one below that is the best fit for your research. If you are not sure, read the appropriate sections before making your selection.

☒ Life sciences ☐ Behavioural & social sciences ☐ Ecological, evolutionary & environmental sciences

For a reference copy of the document with all sections, see [nature.com/documents/nr-reporting-summary-flat.pdf]()

# Life sciences study design

All studies must disclose on these points even when the disclosure is negative.

| | |
|---|---|
| Sample size | No statistical method was used to predetermine sample size. Instead, we made an informed decision based on typical sample sizes used in previous publications employing comparable experimental setups for Arabidopsis (References 10–12), and for experiments in mice (References 21,33,34 ). |
| Data exclusions | No data was excluded in the analysis that fulfilled the set standards and thresholds. During plant experiments, individual plants were excluded before treatment if plant development was not according to standards and previous experiences. All other removals were discussed on an individual basis. |
| Replication | The main proof of concept experiments were performed in a total two replicates, both of which were successful and are included in this study. For all other experiments the number of replicates is stated in the methods section. |
| Randomization | For the plant experiments, treatments were distributed in a regular pattern across the 6-well plates. Mice were assigned to treatment groups at random, but the treatment for each group had to be known at all times in accordance with animal welfare legislation. Samples from different treatments for amplicon sequencing were harvested and processed in random order to avoid batch effects. |
| Blinding | For the plant experiments, the experimenter was fully blind to the allocation of treatments after inoculation. Due to the nature of the mouse experiments, the experimenter could not be blinded to the allocation of treatments. |

# Behavioural & social sciences study design

All studies must disclose on these points even when the disclosure is negative.

| | |
|---|---|
| Study description | N/A |
| Research sample | N/A |
| Sampling strategy | N/A |
| Data collection | N/A |
| Timing | N/A |
| Data exclusions | N/A |
| Non-participation | N/A |

| Randomization | N/A |
|---|---|

# Ecological, evolutionary & environmental sciences study design

All studies must disclose on these points even when the disclosure is negative.

| Study description | N/A |
|---|---|
| Research sample | N/A |
| Sampling strategy | N/A |
| Data collection | N/A |
| Timing and spatial scale | N/A |
| Data exclusions | N/A |
| Reproducibility | N/A |
| Randomization | N/A |
| Blinding | N/A |

Did the study involve field work?  ☐ Yes   ☐ No

## Field work, collection and transport

| Field conditions | N/A |
|---|---|
| Location | N/A |
| Access & import/export | N/A |
| Disturbance | N/A |

# Reporting for specific materials, systems and methods

We require information from authors about some types of materials, experimental systems and methods used in many studies. Here, indicate whether each material, system or method listed is relevant to your study. If you are not sure if a list item applies to your research, read the appropriate section before selecting a response.

## Materials & experimental systems

| n/a | Involved in the study |
|---|---|
| ☒ ☐ | Antibodies |
| ☒ ☐ | Eukaryotic cell lines |
| ☒ ☐ | Palaeontology and archaeology |
| ☐ ☒ | Animals and other organisms |
| ☒ ☐ | Clinical data |
| ☒ ☐ | Dual use research of concern |
| ☐ ☒ | Plants |

## Methods

| n/a | Involved in the study |
|---|---|
| ☒ ☐ | ChIP-seq |
| ☒ ☐ | Flow cytometry |
| ☒ ☐ | MRI-based neuroimaging |

## Antibodies

| Antibodies used | N/A |
|---|---|
| Validation | N/A |

# Eukaryotic cell lines

Policy information about cell lines and Sex and Gender in Research

| | |
|---|---|
| Cell line source(s) | N/A |
| Authentication | N/A |
| Mycoplasma contamination | N/A |
| Commonly misidentified lines (See ICLAC register) | N/A |

# Palaeontology and Archaeology

| | |
|---|---|
| Specimen provenance | N/A |
| Specimen deposition | N/A |
| Dating methods | N/A |

☐ Tick this box to confirm that the raw and calibrated dates are available in the paper or in Supplementary Information.

| | |
|---|---|
| Ethics oversight | N/A |

Note that full information on the approval of the study protocol must also be provided in the manuscript.

# Animals and other research organisms

Policy information about studies involving animals; ARRIVE guidelines recommended for reporting animal research, and Sex and Gender in Research

| | |
|---|---|
| Laboratory animals | Mus musculus C57BL6 with and without Oligo-MM12 microbial community, between 8 and 14 weeks old. |
| Wild animals | No wild animals were used in this study. |
| Reporting on sex | For the animal experiments, male and female 8–14-week-old mice were assigned randomly to experimental groups with no bias for sex. |
| Field-collected samples | No field collected samples were used in the study. |
| Ethics oversight | All animal experiments, including those validating E. coli 8178, S. Tm SL1344, and ATCC14028S, underwent review and approval by the Tierversuchskommission and Kantonales Veterinäramt Zürich. These experiments were conducted under licenses ZH158/2019, ZH108/2022, and ZH109/2022, and followed both cantonal and Swiss legislation. |

Note that full information on the approval of the study protocol must also be provided in the manuscript.

# Clinical data

Policy information about clinical studies
All manuscripts should comply with the ICMJE guidelines for publication of clinical research and a completed CONSORT checklist must be included with all submissions.

| | |
|---|---|
| Clinical trial registration | N/A |
| Study protocol | N/A |
| Data collection | N/A |
| Outcomes | N/A |

# Dual use research of concern

Policy information about dual use research of concern

## Hazards

Could the accidental, deliberate or reckless misuse of agents or technologies generated in the work, or the application of information presented in the manuscript, pose a threat to:

No | Yes

☒ ☐ Public health

☒ ☐ National security

☒ ☐ Crops and/or livestock

☒ ☐ Ecosystems

☒ ☐ Any other significant area

### Experiments of concern

Does the work involve any of these experiments of concern:

No | Yes

☒ ☐ Demonstrate how to render a vaccine ineffective

☒ ☐ Confer resistance to therapeutically useful antibiotics or antiviral agents

☒ ☐ Enhance the virulence of a pathogen or render a nonpathogen virulent

☒ ☐ Increase transmissibility of a pathogen

☒ ☐ Alter the host range of a pathogen

☒ ☐ Enable evasion of diagnostic/detection modalities

☒ ☐ Enable the weaponization of a biological agent or toxin

☒ ☐ Any other potentially harmful combination of experiments and agents

# Plants

| Seed stocks | Arabidopsis thaliana Col0 |
|---|---|
| Novel plant genotypes | *No novel plant genotypes were used in this study.* |
| Authentication | *The genotypes were validated using a multi locus, PCR based approach.* |

# ChIP-seq

## Data deposition

☐ Confirm that both raw and final processed data have been deposited in a public database such as GEO.

☐ Confirm that you have deposited or provided access to graph files (e.g. BED files) for the called peaks.

| Data access links<br>*May remain private before publication.* | N/A |
|---|---|
| Files in database submission | N/A |
| Genome browser session<br>(e.g. UCSC) | N/A |

## Methodology

| Replicates | N/A |
|---|---|
| Sequencing depth | N/A |
| Antibodies | N/A |
| Peak calling parameters | N/A |
| Data quality | N/A |
| Software | N/A |

# Flow Cytometry

## Plots

Confirm that:

☐ The axis labels state the marker and fluorochrome used (e.g. CD4-FITC).

☐ The axis scales are clearly visible. Include numbers along axes only for bottom left plot of group (a 'group' is an analysis of identical markers).

☐ All plots are contour plots with outliers or pseudocolor plots.

☐ A numerical value for number of cells or percentage (with statistics) is provided.

## Methodology

| | |
|---|---|
| Sample preparation | N/A |
| Instrument | N/A |
| Software | N/A |
| Cell population abundance | N/A |
| Gating strategy | N/A |

☐ Tick this box to confirm that a figure exemplifying the gating strategy is provided in the Supplementary Information.

# Magnetic resonance imaging

## Experimental design

| | |
|---|---|
| Design type | N/A |
| Design specifications | N/A |
| Behavioral performance measures | N/A |

## Acquisition

| | |
|---|---|
| Imaging type(s) | N/A |
| Field strength | N/A |
| Sequence & imaging parameters | N/A |
| Area of acquisition | N/A |

Diffusion MRI       ☐ Used       ☐ Not used

## Preprocessing

| | |
|---|---|
| Preprocessing software | N/A |
| Normalization | N/A |
| Normalization template | N/A |
| Noise and artifact removal | N/A |
| Volume censoring | N/A |

## Statistical modeling & inference

| | |
|---|---|
| Model type and settings | N/A |
| Effect(s) tested | N/A |

Specify type of analysis:       ☐ Whole brain       ☐ ROI-based       ☐ Both

Statistic type for inference

(See Eklund et al. 2016)

| N/A |

Correction

| N/A |

## Models & analysis

| n/a | Involved in the study |
|---|---|
| ☐ | ☐ Functional and/or effective connectivity |
| ☐ | ☐ Graph analysis |
| ☐ | ☐ Multivariate modeling or predictive analysis |

Functional and/or effective connectivity

| N/A |

Graph analysis

| N/A |

Multivariate modeling and predictive analysis

| N/A |

