## [Peer Review File · Nature Microbiology]

Peer Review Information

Journal: Nature Microbiology

Manuscript Title: Assessing microbiome population dynamics using wild-type isogenic standardized hybrid (WISH)-tags

Corresponding author name(s): Professor Julia Vorholt

Reviewer Comments & Decisions:

Decision Letter, initial version:

Message: 21st November 2023

Dear Julia,

Thank you for your patience while your manuscript "Wild-type Isogenic Standardized Hybrid (WISH)-tags for assessing population dynamics within microbiomes" was under peer-review at Nature Microbiology. It has now been seen by 4 referees, whose expertise and comments you will find at the of this email. You will see from their comments below that while they find your work of interest, some important points are raised. We are very interested in the possibility of publishing your study in Nature Microbiology, but would like to consider your response to these concerns in the form of a revised manuscript before we make a final decision on publication.

In particular, we ask that you address all concerns with text edits including a detailed discussion of limitations and please clearly clarify the advance (methodological and biological). We will not require you to perform additional experiments to provide biological insight, however, if you already have these data, we would strongly encourage you to add it. The rest referees' reports are clear and the remaining issues should be straightforward to address. We are overruling concerns with novelty.

If you have not done so already please begin to revise your manuscript so that it conforms to our Resource format instructions at <http://www.nature.com/nmicrobiol/info/final-submission/>

The usual length limit for a Nature Microbiology Article is six display items (figures or tables) and 3,000 words. We have some flexibility, and can allow a revised manuscript at 3,500 words, but please consider this a firm upper limit. There is a trade-off of ~250 words per display item, so if you need more space, you could move a Figure or Table to Supplementary Information.

Some reduction could be achieved by focusing any introductory material and moving it to the start of your opening 'bold' paragraph, whose function is to outline the background to your work, describe in a sentence your new observations, and explain your main conclusions. The discussion should also be limited. Methods should be described in a separate section following the discussion, we do not place a word limit on Methods.

Nature Microbiology titles should give a sense of the main new findings of a manuscript, and should not contain punctuation. Please keep in mind that we strongly discourage active verbs in titles, and that they should ideally fit within 90 characters each (including spaces).

Please include a data availability statement as a separate section after Methods but before references, under the heading "Data Availability". This section should inform readers about the availability of the data used to support the conclusions of your study. This information includes accession codes to public repositories (data banks for protein, DNA or RNA sequences, microarray, proteomics data etc...), references to source data published alongside the paper, unique identifiers such as URLs to data repository entries, or data set DOIs, and any other statement about data availability. At a minimum, you should include the following statement: "The data that support the findings of this study are available from the corresponding author upon request", mentioning any restrictions on availability. If DOIs are provided, we also strongly encourage including these in the Reference list (authors, title, publisher (repository name), identifier, year). For more guidance on how to write this section please see: <http://www.nature.com/authors/policies/data/data-availability-statements-data-citations.pdf>

To improve the accessibility of your paper to readers from other research areas, please pay particular attention to the wording of the paper's opening bold paragraph, which serves both as an introduction and as a brief, non-technical summary in about 150 words. If, however, you require one or two extra sentences to explain your work clearly, please include them even if

the paragraph is over-length as a result. The opening paragraph should not contain references. Because scientists from other sub-disciplines will be interested in your results and their implications, it is important to explain essential but specialised terms concisely. We suggest you show your summary paragraph to colleagues in other fields to uncover any problematic concepts.

If your paper is accepted for publication, we will edit your display items electronically so they conform to our house style and will reproduce clearly in print. If necessary, we will re-size figures to fit single or double column width. If your figures contain several parts, the parts should form a neat rectangle when assembled. Choosing the right electronic format at this stage will speed up the processing of your paper and give the best possible results in print. We would like the figures to be supplied as vector files - EPS, PDF, AI or postscript (PS) file formats (not raster or bitmap files), preferably generated with vector-graphics software (Adobe Illustrator for example). Please try to ensure that all figures are non-flattened and fully editable. All images should be at least 300 dpi resolution (when figures are scaled to approximately the size that they are to be printed at) and in RGB colour format. Please do not submit Jpeg or flattened TIFF files. Please see also 'Guidelines for Electronic Submission of Figures' at the end of this letter for further detail.

Figure legends must provide a brief description of the figure and the symbols used, within 350 words, including definitions of any error bars employed in the figures.

When submitting the revised version of your manuscript, please pay close attention to our <https://www.nature.com/nature-research/editorial-policies/image-integrity> Digital Image Integrity Guidelines. and to the following points below:

Please include a statement before the acknowledgements naming the author to whom correspondence and requests for materials should be addressed.

Finally, we require authors to include a statement of their individual contributions to the paper -- such as experimental work, project planning, data analysis, etc. -- immediately after the acknowledgements. The statement should be short, and refer to authors by their initials. For

details please see the Authorship section of our joint Editorial policies at http://www.nature.com/authors/editorial_policies/authorship.html

- * include a point-by-point response to any editorial suggestions and to our referees. Please include your response to the editorial suggestions in your cover letter, and please upload your response to the referees as a separate document.

- * ensure it complies with our format requirements for Letters as set out in our guide to authors at www.nature.com/nmicrobiol/info/gta/

- * state in a cover note the length of the text, methods and legends; the number of references; number and estimated final size of figures and tables

*This url links to your confidential homepage and associated information about manuscripts you may have submitted or be reviewing for us. If you wish to forward this e-mail to co-authors, please delete this link to your homepage first.

Please ensure that all correspondence is marked with your Nature Microbiology reference number in the subject line.

Nature Microbiology is committed to improving transparency in authorship. As part of our efforts in this direction, we are now requesting that all authors identified as 'corresponding author' on published papers create and link their Open Researcher and Contributor Identifier (ORCID) with their account on the Manuscript Tracking System (MTS), prior to acceptance. This applies to primary research papers only. ORCID helps the scientific community achieve unambiguous attribution of all scholarly contributions. You can create and link your ORCID from the home page of the MTS by clicking on 'Modify my Springer Nature account'. For more information please visit please visit www.springernature.com/orcid.

We hope to receive your revised paper within three weeks. If you cannot send it within this time, please let us know.

Yours sincerely,

Reviewer Expertise:

Referee #1: plant microbiome, omics

Referee #2: mammalian microbiomes, sequencing based methods, bacterial pathogenesis

Referee #3: omics, microbiome

Referee #4: mammalian microbiomes, ecology

Reviewers Comments:

Reviewer #1 (Remarks to the Author):

This paper aims to create a universal genomic tagging system for bacteria that does not negatively affect their fitness. It largely takes advantage of the Tn7 transposon, a mobile element that is known to insert downstream of the *glmS* gene in many bacteria in a non-essential region without major fitness effects. The authors design their tags to be robust to the common detection methods of qPCR and amplicon sequencing. They insert their tags into a variety of bacteria and do a couple small experiments that demonstrate the kind of information that tags allow – namely how isogenic strains behave with each other. For those bacteria that cannot be conveniently tagged using a Tn7 system, they use homologous recombination.

The idea of tagging bacteria in some way with Tn7 is not unique; this paper builds on a large body of existing work. What is unique in this paper is the clever design of the barcodes. By using two 20 bp tags, they construct unique sequences that can be distinguished, of course, by amplicon sequencing, but also using qPCR primers. They pre-screened their tags to avoid sequences that are likely to be found in bacteria, thus helping ensure that the only qPCR amplicons that can be produced will have arisen from the intended barcoded region. I find this a very nice development that allows one to use either or both methods. Of note, qPCR can quantify absolute abundances whereas traditional amplicon sequencing fails to do that, so it is especially nice to include that as an option.

The experiments done on priority effects are quite elegantly constructed. The authors are honest about the limitations and extrapolation possibilities from their limited-scale tests, so I don't have any problem there.

Line 223-224 "metaphorically described to carry their guts on the outside".

>> This metaphor refers to plant roots in every situation I have seen it used, and only makes sense to me in that context because it is in the roots that there is a lot of water and nutrient exchange. However, the following line, and all the experiments, goes on to discuss the *A. thaliana* phyllosphere, for which this metaphor is no longer relevant... I suggest removing this metaphor or rewriting to clarify.

Line 357 "The ... sequences were generated with 19 bp ACGT followed by 3 bp AT..."

>> I think the authors are saying 19 bases is any base A, C, T, or G, followed by 3 bp A or T, etc. I think it should be written more clearly like this. The phrase "19 bp ACGT" has no clear meaning of its own and leaves one to guess. Most may guess right, but better to just be clear.

For the amplicon sequencing, the authors suggest using an amplicon of 120 in addition to the length of the overhangs. This length works well for the PCR protocol they have chosen.. that is, using phosphatase destroy the first round of primers before adding the primers that include Illumina adapters. However, other labs may for example prefer to use SPRI beads (eg. Ampure XP) for primer cleanup, and a shorter amplicon may prove problematic as it can be lost to some degree during cleanup. Further, there are interesting amplicon-seq protocols that involve multiple primers (eg. <https://elifesciences.org/articles/66186>) that need to be removed by SPRI beads, and a short amplicon may not be compatible with these. The full inserted WISH tag cassette includes a KanR gene and is longer than 120 (or so it appears in Supplementary Fig 17)... can the authors suggest any other priming site if users prefer to increase their amplicon size for whatever reason, or at least add some discussion about this?

Fig 1a: The 40 bp barcode consists of a 16bp region and a 24bp region. Perhaps these subsections could be indicated directly in the orange with a clear annotation. I see that the authors have made one of the orange rectangles bigger than the other with a tiny break in between the two boxes, but this is quite subtle in my opinion.

>> Neither the word "conjugation" nor "electroporation" appears in the methods. It is unclear how the authors transformed their microbes. A table in the supplement listing conditions used for each of the microbes mentioned (electroporation or conjugation, parameters, antibiotic concentration for selection, etc.) should be included.

>> In Supplementary Figure 2 I guess BPM in the chart legend means "bacterial plant mouse"... the figure legend can make this clear. Also, the "dashed grey line" that represents the detection limit is hard to distinguish from the grey boxes, that actually look a lot like their own like a "dashed grey line" but mean something totally different.

>> In Supplementary figure 1, it refers to "Levenshtein distance"... I wasn't familiar with and it wasn't defined, but according to Wikipedia this is often used to mean "the minimum number of single-character edits required to change one word into the other.". Is the authors meaning equally simple here? If so, the legend should also explain it in plain words, because the concept is then actually simple and calling it the "Levenshtein distance" just confuses readers like me.

Line 235: " As the plants grew considerably between 0 and 7 dpi, and also continuously produced new leaves, the volume of the inoculum was adapted according to the size of the plant (keeping the inoculation population relative to the plant weight constant)."

>> How? This was on a plant-by-plant basis, or was done by averages? Did I miss it? Can it

be made more clear?

Line 485: Source of calcined clay?

Reviewer #2 (Remarks to the Author):

In "WISH-tags for assessing population dynamics within microbiomes", Daniel et al present a method to improve studies of microbial population dynamics; however, the nature of the innovation is not clear. Previous studies from this group and others (e.g. refs 18, 19) have used barcodes in very similar fashions. They claim that the capacity to do both qPCR and sequencing on WISH-tags is a major development, but as demonstrated by nearly all the data presented in this paper, the need for qPCR is not made clear. Also, designing primers (for qPCR) corresponding to barcode sequences is not a major innovation and any library that can be analyzed by sequencing could, theoretically, also be analyzed by qPCR.

The applications of WISH-tags to murine gut and plant microbiome investigations are thorough and well-controlled, but do not uncover new biology. Priority effects in both models have been studied. Moreover, WISH-tags are not required to investigate priority effects, so using super-colonization resistance as a demonstration of WISH-tag utility does not reveal their potential power. If they want to claim that WISH-tags open new avenues for discovery of new biology they should show it. For example, the authors claim that WISH-tags will be useful in discovering novel bottleneck biology. However, if the bottleneck being measured is relatively permissive, 62 tags will not impart sufficient resolution to measure the contraction of the population.

Minor issues

1. Within the text many of the citations of the supp material are incorrect.
2. Similarly, within the figure legends, the description of the images is inaccurate (e.g. there is no 'gray' in Fig 3b)
3. Summary statistics are not given in any of the Figure legends (e.g. number of observations, measure of center, etc.).
4. The supplementary figures are often not adequately described (e.g. Supp 12 panel 1-q)
5. the imputed CFU from the barcode fraction is questionable at best and should be specifically described.
6. The tags themselves are not 'isogenic' making the title misleading.
7. The accession number is not available in the methods.

Reviewer #3 (Remarks to the Author):

This manuscript describes a method for incorporating standardized, phenotype-neutral barcode sequences into cultivated microbes for tracking population changes with either qPCR

7or sequencing. They incorporate these barcodes into strains of the gut-colonizing bacteria *Salmonella enterica* and *E. coli* as well as multiple leaf-colonizing bacteria, and use the readouts to demonstrate differences in microbiome resistance to invasion. They observe that the gut microbiome is largely resistant to colonization to late-arriving strains while the leaf microbiome is relatively permissive.

Overall this is a nice study. The methodological advance is straightforward, built on existing approaches but creating a nice system for a standardized and flexible approach that could be used for multiple microorganisms (although even within the strains tested, some troubleshooting and adaptation was necessary). The observations on colonization are not a huge surprise given current knowledge of gut and leaf microbiomes, but allow for a cleaner and more precise quantification of the effects.

Detailed comments:

Line 33: I found "Microorganisms are ubiquitous in terrestrial and marine ecosystems" to be a poor choice of introductory sentence for a study exclusively focused on host-associated microbiomes. I understand the authors want to make the point that their approach could potentially be applied to a wide variety of environments, but maybe they could start out with something a little more relevant to the study?

Line 39: "Microbial interactions between taxa are typically revealed by co-occurrence networks based on 16S/18S rRNA sequencing..." I think there are many ways of characterizing microbial interactions that go beyond co-occurrence networks. And why is this relevant to the study here that does not investigate pairwise interactions?

Line 55: "RB-Seq" should be "RB-Tn-Seq" (for Reverse Barcoded Transposon-site sequencing).

Line 78: Why are the tags labeled as "wild type", which to me would mean they are derived from nature and that is not the case for either the sequence tags or the tagged strains?

Supplementary Figure 1 presents the Levenshtein distances between tags. However the methods section does not describe any step in the design of the tags that ensures appropriate distance – they are described as "random strings of bases" (line 248) that are filtered for GC content, palindromic sequences, alignment to host genomes, runs of identical bases and melting temperatures (lines 248-352). How was appropriate distance achieved?

Line 122: "...mice raised in a pathogen free environment..." Was it not a germ-free environment? The same mice are referred to later in the manuscript as "Germ-free (GF) mice" (line 193).

Line 123: "From now on, we will refer to this pooled DNA as BPM-DNA." It would be helpful to explain the reason for calling the pooled background DNA "BPM-DNA" – it only became apparent later that this must stand for "Bacterial – Plant – Mouse". Also, in the methods section there is a similar description of creating a mixed DNA pool, along with the statement, "This mix, from now on referred to as background DNA..." (line 376). Is this not the same thing? If they are please be consistent with the naming.

Line 185: Here the manuscript says that fecal pellets were collected for microbiota assessment, noting that "fecal *S. Tm* populations recapitulate the *S. Tm* population in the cecum". But the methods section on "DNA extraction mouse feces" describes overnight cultivation prior to genomic DNA extraction (lines 591-595) – does this recapitulate the cecum populations?

Line 243: Here and elsewhere the reader is referred to Supplementary Figure 10 for details on experimental methods, but Supplementary Figure 10 displays Lipocalin-2 values. I believe the figure that should be referenced is Supplementary Figure 8.

Similarly, in the paragraph starting on line 244, readers are pointed to Supplementary Figure 13 for data on colonization density in the context of the background community, which looks like it is actually in Supplementary Figure 11, and to Supplementary Figure 14q, which does not exist, for the statement "The carrying capacity remained the same as in the wild type Leaf257" which I don't fully understand but I think is supposed to be represented in Supplementary Figure 12 (which does have a panel q). Overall, the overwhelming number of supplementary figures and their poor referencing in the text tend to detract from the message rather than enhance it.

Line 282: The manuscript recommends the mBARq tool and references a submitted manuscript. I appreciate that this was provided to the reviewers, but maybe a pointer to a site where the software is available would be more helpful to readers particularly if the other manuscript isn't published yet?

Lines 410 and 415 (and others): One of the Salmonella strains is described as both "ATCC1428S" and "ATCC14028S" – I assume these are the same strain and only one of the names is correct? Also, on lines 471-475 it is noted that one barcoded strain of this genotype consistently failed to colonize and is assumed to "be afflicted by an off-target effect which reduces its fitness in the inflamed gut." But was this barcoded strain discarded or still used in the experiments?

Line 522: "...samples were collected which were spotted on selective and non-selective R2A square plates to determine CFU and wild-type to barcoded strain ratio." How was this ratio determined?

Lines 532 and 537: "Mouse AW229 was removed from analysis because of clearing of all S. Tm from the gut for unknown reasons." And "Eight mice in the GF Early-Late, one of which was removed from the downstream analysis because of clearing of S. tm." Do these both refer to the same mouse that was removed? Also the second is an incomplete sentence.

Line 574: The manuscript says some plants were not harvested because of abnormal phenotypes, clarifying "meaning they were either very small.." but there is no "or" to continue this explanation.

Line 583: The entire plant was completely submerged in 600 uL buffer? Weren't some of these plants 7 or 14 days old?

Line 597: Why was 400 uL of Leaf257 culture added to "plant wash-off" DNA extractions?

Line 628: The data availability statement lacks an accession number. Have the data been deposited?

Figure 1: It would be helpful if lengths of all parts of the tags we included in panel a. Panel C as well.

Figure 3: What is the "grey area" referred to in panel B?

Figure 4: This legend points to Supplementary Figure 17 for tag abundance data, but that figure contains cloning procedures; I think Supplementary Figure 15 contains the relevant information.

Supplementary Figure 15: the legend describes "...the right half with the grey background..." but there are grey panels interspersed with white panels and it's not readily clear to me what

is what.

Minor comments:

Italicization is inconsistent for terms like *in vitro*, *in situ*, *in planta*, etc. – either they should all be italicized or none.

Lines 147-148: It was hard to follow this sentence and understand that the three *Salmonella* strains are ATCC14028S, SL1344 (SB300), both of which are commonly used in enteric bacteria research, and an avirulent mutant of SL1344.

Line 315: Should be “priority effect” not “priority affect”.

Line 327: Consider rephrasing “...can be used in specific but also across microbiomes...”

Line 488: There is an extra close parenthesis

Line 544: Incomplete sentence

Reviewer #4 (Remarks to the Author):

This work by Daniel et al. introduces a new genome tagging system designed for the tracking the abundance of isogenic strains in complex microbial communities. The approach is straightforward and is compatible with qPCR and Illumina sequencing making it applicable in diverse scenarios. The authors validated the quantitative aspects of their approach, demonstrating its capability to span 5 orders of abundance magnitude. As a proof of concept, the approach was tested on a murine and plant system, revealing a priority effect in mice but not in the *Arabidopsis* phyllosphere.

The strengths of this study include the ability to discriminate, track, and characterize closely related strains of the same species within a complex native microbiota in a reliable, quantitative manner. The weaknesses of the study include a requirement for isolation of the strain under laboratory culture conditions, and low throughput tagging of these isolates. The demonstration experiments were performed under conditions of limited complexity, and the biological findings from these experiments are somewhat expected and do not advance our understanding of these host-microbiota ecosystems in a substantive manner. Overall, the contributions of this paper are more strongly oriented towards methods than towards biology.

Major points

1. Consider explaining in the Discussion the limitations on the species that can be tagged. For example, what are the implications of requiring laboratory growth? How many of the numerically most abundant native strains from each of these experimental systems are reliably cultivated and how representative are the cultivated strains of each of these species? (*S. Typhimurium* and *E. coli* are not common commensals of *Mus musculus*.)

2. How conserved is the *glmS* gene in the indigenous microbiota of these model systems? How

10important is this gene to the proposed strategy? How common are Tn7 transposase sites close to glmS? Of note, the authors report that several sites had to be tested in the Rhizobium strain of interest, before an appropriate one could be found. This suggests that the process of generating tagged strains may be quite laborious.

3. The discussion can benefit from addressing genetic changes that can in theory occur during the tagging process and/or after colonization. Consider sharing the number of generations that it takes to tag a strain. This comment is related to what seem to be pre- and post- inoculation differences shown in Supp. Figure 9.

4. Further discussion would be useful about the advantages of this method over state-of-the-art strain-resolved metagenomic assembly methods.

5. The authors should make clear what new information about priority effects was learned in these experiments that was either not known previously or could not have been learned using existing methods? How is this method superior to state-of-the-art strain-resolved metagenomic assembly methods?

Minor points

6. Consider specifying in that abstract that exactly two species were tested for a priority effect. As written, it is not clear if Salmonella and Sphingomonas were the only species assessed.

7. Line 20. The claim of no background DNA amplification should be modified. More precisely, there was no background DNA detected above the limits of detection. The claim should be revised to reflect this, and ideally state the limits of detection. Similarly in line 98 and elsewhere, the claim that background DNA was "excluded" should be revised.

8. Line 47. The claim that barcodes are ideal for tracing population composition is over-stated. They have advantages and disadvantages. Among the latter, they fail to allow for monitoring extrachromosomal elements, and other forms of genome evolution.

9. Line ~50. Contrast with in vivo murine work like Vasquez et al. Cell Host Microbe 2021, 10.1016/j.chom.2021.08.003.

10. Line 57. In theory, no insertion into a genome is perfectly neutral. What the authors may mean by neutral is the absence of a detected fitness effect. As with any negative finding, it may be important to qualify the finding with the sensitivity of the measurement method.

11. Line 63. Gnotobiotic animal models differ from naturally colonized animals of the same host lineage in more than the absence of microbiota. Having been born to a germ-free dam/mother, they fail to complete post-natal tissue and organ differentiation. They have very different host physiology at birth, and even after mono- or more complex colonization, host

development follows a different trajectory. To the degree that host biology and immunology control microbiota colonization, gnotobiotic animals provide a somewhat aberrant view of microbiota assembly.

12. Line 99. To avoid confusion please correct “the only difference between the tags” -> “tags differed in a 40bp region” or similar.

13. Figure 2a. Did you add ~100k molecules per tag? If you did, you should specify that in the legend. In the current wording, you specify only “equal concentrations”. Also – please state that the dashed horizontal line is the detection limit, and either define it here in legends or refer to a definition elsewhere (e.g., in supp. fig. 2).

14. Figure 2b. Please define “normalized counts”.

15. Figure 2b. Why are there so many outliers at dilution 7? Getting suddenly 1000 reads when you added close to nothing seems strange.

16. Figure 2c. Did you choose not to show dilutions 6 and 7? If so, please state such and consider expanding in the legends or main text on NGS issues when the number of molecules is very small or zero.

17. Line 149. “were barcoded” – with how many barcodes? Reading further it seems that “six barcodes each” on line 151 may refer to all strains. Please clarify.

18. Line 153. Not clear for an outsider why you need a different integration system. If possible, explain briefly. Also explain here or in the discussion which bacteria are suitable for this approach, and which are not. (see Comment #)

19. Line 158. Please add a separate statement or sentence for all supplementary figures 5 to 9.

20. Supp. Figure 9. Looks like the tagged strains might not be genetically identical. For example, WISH-25 is expanding while WISH-27 is declining in the 4 mice in panel a, oligo-MM. Would you assess these differences to be beyond neutral drift? Perhaps add comment in discussion about the possibility of in vivo evolution or other sources of genetic variance such as pre-inoculation differences.

21. Supp. Figure 11. “The y-axis shows the time and the x-axis the CFU per plant.” The authors have mixed up the x- and y-axes in this statement. All legends to Supplementary Figures should be carefully checked for wording and errors.

22. Around line 190. Did you use shotgun or qPCR here? Please be explicit about the approach taken.

23. Line 206 and Figure 3b. I can't clearly see a pink line (I am slightly color blind), a gray area, or arrows. Please enhance figure quality and perhaps make colors more pronounced if needed.
24. Figure 3c. I didn't see a reference from the main text to Figure 3c.
25. Line 250. Is this reference to "Supp. Figure 14q" intended to refer to "Supp. Figure 12q"?
26. Line 254. "Supplementary Figure 16be" seems to be a typo.
27. Line 261. "Supplementary Figure 17" may also be a typo.
28. Line 407. Allowing up to 5 errors seems unnecessary since a typical sequencing error rate is only 0.5% per bp. Do you think this might explain the false positives in Figure 2c, dilution 7?
29. Line 424. Please refer to Supp. Figure 17 from the main text. Could be informative for non-experts to hear about the integration process in a single sentence.

Author Rebuttal to Initial comments

We thank the reviewers for taking the time to review our manuscript and their valuable comments.

Reviewer Expertise:

Referee #1: plant microbiome, omics

Referee #2: mammalian microbiomes, sequencing based methods, bacterial pathogenesis

Referee #3: omics, microbiome

Referee #4: mammalian microbiomes, ecology

Reviewers Comments:

Reviewer #1 (Remarks to the Author):

This paper aims to create a universal genomic tagging system for bacteria that does not negatively affect their fitness. It largely takes advantage of the Tn7 transposon, a mobile element that is known to insert

13downstream of the *glmS* gene in many bacteria in a non-essential region without major fitness effects. The authors design their tags to be robust to the common detection methods of qPCR and amplicon sequencing. They insert their tags into a variety of bacteria and do a couple small experiments that demonstrate the kind of information that tags allow – namely how isogenic strains behave with each other. For those bacteria that cannot be conveniently tagged using a Tn7 system, they use homologous recombination.

The idea of tagging bacteria in some way with Tn7 is not unique; this paper builds on a large body of existing work. What is unique in this paper is the clever design of the barcodes. By using two 20 bp tags, they construct unique sequences that can be distinguished, of course, by amplicon sequencing, but also using qPCR primers. They pre-screened their tags to avoid sequences that are likely to be found in bacteria, thus helping ensure that the only qPCR amplicons that can be produced will have arisen from the intended barcoded region. I find this a very nice development that allows one to use either or both methods. Of note, qPCR can quantify absolute abundances whereas traditional amplicon sequencing fails to do that, so it is especially nice to include that as an option.

The experiments done on priority effects are quite elegantly constructed. The authors are honest about the limitations and extrapolation possibilities from their limited-scale tests, so I don't have any problem there.

Line 223-224 “metaphorically described to carry their guts on the outside”.

>> This metaphor refers to plant roots in every situation I have seen it used, and only makes sense to me in that context because it is in the roots that there is a lot of water and nutrient exchange. However, the following line, and all the experiments, goes on to discuss the *A. thaliana* phyllosphere, for which this metaphor is no longer relevant... I suggest removing this metaphor or rewriting to clarify.

We have removed the metaphor as suggested.

Line 357 “The ... sequences were generated with 19 bp ACGT followed by 3 bp AT...”

>> I think the authors are saying 19 bases is any base A, C, T, or G, followed by 3 bp A or T, etc. I think it

should be written more clearly like this. The phrase “19 bp ACGT” has no clear meaning of its own and leaves one to guess. Most may guess right, but better to just be clear.

We revised the description. It now reads: “The 24 bp sequences were generated with the first 19 bp taken from any of the four bases, followed by 3 bp A or T and then 2 bp G or C, making the ...”

For the amplicon sequencing, the authors suggest using an amplicon of 120 in addition to the length of the overhangs. This length works well for the PCR protocol they have chosen.. that is, using phosphatase destroy the first round of primers before adding the primers that include Illumina adapters. However, other labs may for example prefer to use SPRI beads (eg. Ampure XP) for primer cleanup, and a shorter amplicon may prove problematic as it can be lost to some degree during cleanup. Further, there are interesting amplicon-seq protocols that involve multiple primers (eg. <https://elifesciences.org/articles/66186>) that need to be removed by SPRI beads, and a short amplicon may not be compatible with these. The full inserted WISH tag cassette includes a KanR gene and is longer than 120 (or so it appears in Supplementary Fig 17)... can the authors suggest any other priming site if users prefer to increase their amplicon size for whatever reason, or at least add some discussion about this?

Regarding the bead clean-up: We performed a bead clean-up as the final step to prepare the libraries for sequencing. At this point, the length of the fragment has increased to 262 bp and we did not observe a major loss in material. After the first round of amplifications, the fragment is 186 bp long and should already be long enough for successful bead clean-up. We had tested the described protocol also with smaller fragments (using a commercial ladder), even cleanup of fragments of only 70 bp were successful. We did not test Ampure XP but expect similar efficiency.

Regarding alternative priming sites: Alternative resistance cassettes can be used in conjunction with the WISH tags. However, in our study, we primarily used the KanR cassette. We have adapted the caption of Supplementary Figure 17 to indicate that other antibiotic resistances can also be used. Nevertheless, for the *At*-LSPHERE strains all our integrations contain one potential priming site that would result in a 188 bp amplicon excluding the overhangs using the universal forward primer. Including the overhangs from the first PCR, this would extend to 254 bp. Primers like those described in the cited article should be removable without significant loss of amplicon using a higher proportion of beads to sample.

Fig 1a: The 40 bp barcode consists of a 16bp region and a 24bp region. Perhaps these subsections could be indicated directly in the orange with a clear annotation. I see that the authors have made one of the orange rectangles bigger than the other with a tiny break in between the two boxes, but this is quite subtle in my opinion.

We have added a black line to replace the break between the two distinct parts for better visibility.

>> Neither the word “conjugation” nor “electroporation” appears in the methods. It is unclear how the authors transformed their microbes. A table in the supplement listing conditions used for each of the microbes mentioned (electroporation or conjugation, parameters, antibiotic concentration for selection, etc.) should be included.

We have generated a table with the transformation methods and included it in the supplementary_data_files (Supplementary data file 1). For published methods, the original source is cited. In case we adjusted the protocol, we now specify the protocol and cite the respective publication from which the protocol was adapted.

>> In Supplementary Figure 2 I guess BPM in the chart legend means “bacterial plant mouse” ... the figure legend can make this clear. Also, the “dashed grey line” that represents the detection limit is hard to distinguish from the grey boxes, that actually like a lot on their own like a “dashed grey line” but mean something totally different.

We added the abbreviation BPM to the figure caption. We modified the figure and adjusted the color of the points for better visibility.

>> In Supplementary figure 1, it refers to “Levenshtein distance” ... I wasn’t familiar with and it wasn’t defined, but according to Wikipedia this is often used to mean “the minimum number of single-character edits required to change one word into the other.”. Is the authors meaning equally simple here? If so, the legend should also explain it in plain words, because the concept is then actually simple and calling it the “Levenshtein distance” just confuses readers like me.

The meaning is correct and showcases the robustness of the barcodes against mutations, amplification, and sequencing errors. We added this information to the revised figure caption. It now reads: “The Levenshtein distances, that is, the minimal number of base exchanges required in one barcode to turn it into another, of WISH-tag sequences are compared, ...”

Line 235: “ As the plants grew considerably between 0 and 7 dpi, and also continuously produced new leaves, the volume of the inoculum was adapted according to the size of the plant (keeping the inoculation population relative to the plant weight constant).”

>> How? This was on a plant-by-plant basis, or was done by averages? Did I miss it? Can it be made more clear?

The exact volumes used in both timepoints are stated in the methods section (see paragraph relabeled “Sequential arrival of bacterial populations in plant experiments”). The volumes for all inoculations were standardized for both inoculation time points, i.e. 4 and 20 μ L respectively. These were based on the approximately 5-fold increase in mean plant weight. The plant weights at the different timepoints are shown in Supplementary Figure 16. The sentence was rephrased to improve clarity and refers to the Materials and methods section.

Line 485: Source of calcined clay?

The supplier is Cremonini terre rosse (It). The information was added to the methods section.

Reviewer #2 (Remarks to the Author):

In “WISH-tags for assessing population dynamics within microbiomes”, Daniel et al present a method to improve studies of microbial population dynamics; however, the nature of the innovation is not clear. Previous studies from this group and others (e.g. refs 18, 19) have used barcodes in very similar fashions. They claim that the capacity to do both qPCR and sequencing on WISH-tags is a major development, but as demonstrated by nearly all the data presented in this paper, the need for qPCR is not made clear. Also, designing primers (for qPCR) corresponding to barcode sequences is not a major innovation and any library that can be analyzed by sequencing could, theoretically, also be analyzed by qPCR.

The applications of WISH-tags to murine gut and plant microbiome investigations are thorough and well-controlled, but do not uncover new biology. Priority effects in both models have been studied. Moreover, WISH-tags are not required to investigate priority effects, so using super-colonization resistance as a demonstration of WISH-tag utility does not reveal their potential power. If they want to claim that WISH-tags open new avenues for discovery of new biology they should show it. For example, the authors claim that WISH-tags will be useful in discovering novel bottleneck biology. However, if the bottleneck being measured is relatively permissive, 62 tags will not impart sufficient resolution to measure the contraction of the population.

We agree that priority effects have been studied, at least in the murine gut, and we chose them precisely for that reason. It allowed us to have a reasonable comparison to the results we received with the WISH tag barcode system. In the context of plant microbiomes, priority effects within populations of strains have not yet been investigated. In our study, we show that the outcomes of late arrival of populations are clearly distinct between the host systems, revealing that the host environment impacts population dynamics. Our aim was to introduce a tagging system that is benchmarked across multiple microbiomes to improve standardization in experimental approaches.

Regarding the point that any sequencing library can theoretically be analyzed also by qPCR. We optimized the design to avoid common problems such as unequal amplification or differential primer binding, which can introduce biases. To address the point that the advantage of qPCR was not sufficiently made clear, we rephrased the main text line 95, which now reads: "We aimed to develop a system that allows the quantification of barcoded strain populations by both qPCR and NGS for more flexibility. qPCR offers an unparalleled dynamic range, allowing the quantification of abundant and rare barcodes from the same sample, while having a low turnaround time for small sample sizes. On the other hand, NGS-based quantification facilitates a higher throughput of otherwise labor-intensive experiments, while being more economical for large sample numbers."

On the point of the detection of permissive bottlenecks, it is correct that a mild bottleneck will not be detectable with any number of barcodes, while a few might be enough for severe ones, as demonstrated before (Grant et al., 2008). While we empirically validated 62 WISH tags, we provide 500 WISH-tag sequences in the supplementary material and a link that provides access to even more.

Minor issues

1. Within the text many of the citations of the supp material are incorrect.

Thank you for pointing this out. All figures past Supplementary Figure 7 were accidentally shifted by two numbers. We addressed this error in the revised version of the manuscript.

2. Similarly, within the figure legends, the description of the images is inaccurate (e.g. there is no 'gray' in Fig 3b)

This oversight was correct. Thank you.

3. Summary statistics are not given in any of the Figure legends (e.g. number of observations, measure of center, etc.).

We added the information to the figure captions.

4. The supplementary figures are often not adequately described (e.g. Supp 12 panel 1-q)

We improved the figure legends to convey the description of the data more clearly.

5. the imputed CFU from the barcode fraction is questionable at best and should be specifically described.

We improved the presentation of the imputed CFU, which is derived from the relative abundance of the WISH-tagged strains from the sequencing data and applied to the CFU numbers. We display imputed data in Figures 3 b and 4 b to visually convey the population dynamics. Figures 3 c and 4 c show the fractions directly obtained from the sequencing data. We added a description of the imputation to the methods section (line 640 of the revised manuscript).

6. The tags themselves are not 'isogenic' making the title misleading.

The point of reference to call the barcodes isogenic is that the strains are functionally identical, only differing in the 40 bp of the barcode. We rephrased the text to provide this information more clearly. "The newly developed WISH-tags differed only in their core, the 40 bp of the unique barcode region, which ensures sufficient distinctiveness between any two tags". The use appears common, and we adopted it, e.g.

<https://doi.org/10.1371/journal.pbio.0060074>

https://doi.org/10.1007/978-1-4939-9570-7_13

<https://doi.org/10.1007/s00239-023-10103-6>

7. The accession number is not available in the methods.

The accession number PRJEB66333 was added to the manuscript.

Reviewer #3 (Remarks to the Author):

This manuscript describes a method for incorporating standardized, phenotype-neutral barcode sequences into cultivated microbes for tracking population changes with either qPCR or sequencing. They incorporate these barcodes into strains of the gut-colonizing bacteria *Salmonella enterica* and *E. coli* as well as multiple leaf-colonizing bacteria, and use the readouts to demonstrate differences in microbiome resistance to invasion. They observe that the gut microbiome is largely resistant to colonization to late-arriving strains while the leaf microbiome is relatively permissive. Overall this is a nice study. The methodological advance is straightforward, built on existing approaches but creating a nice system for a standardized and flexible approach that could be used for multiple microorganisms (although even within the strains tested, some troubleshooting and adaptation was necessary). The observations on colonization are not a huge surprise given current knowledge of gut and leaf microbiomes, but allow for a cleaner and more precise quantification of the effects.

Detailed comments:

Line 33: I found “Microorganisms are ubiquitous in terrestrial and marine ecosystems” to be a poor choice of introductory sentence for a study exclusively focused on host-associated microbiomes. I understand the authors want to make the point that their approach could potentially be applied to a wide variety of environments, but maybe they could start out with something a little more relevant to the study?

We adjusted the first paragraph to avoid a disconnect. We would still prefer to start with microbiomes in general, but then refer to host microbiomes within the same paragraph.

Line 39: “Microbial interactions between taxa are typically revealed by co-occurrence networks based on 16S/18S rRNA sequencing...” I think there are many ways of characterizing microbial interactions that go beyond co-occurrence networks. And why is this relevant to the study here that does not investigate pairwise interactions?

We agree that our phrasing distracted from the main point of solving population dynamics when we mentioned means by which interactions can occur. We have amended the text to reflect this point.

Line 55: “RB-Seq” should be “RB-Tn-Seq” (for Reverse Barcoded Transposon-site sequencing).

The correction was implemented.

Line 78: Why are the tags labeled as “wild type”, which to me would mean they are derived from nature and that is not the case for either the sequence tags or the tagged strains?

21The name “Wild-type isogenic” is supposed to be taken together to refer to the use of the tags in strains that are functionally unaltered but distinguishable by a barcode. However, the tags can also be used to differentiate mutants, for example to quantify fitness factors, as shown in Vasquez et al., 2021.

Supplementary Figure 1 presents the Levenshtein distances between tags. However the methods section does not describe any step in the design of the tags that ensures appropriate distance – they are described as “random strings of bases” (line 248) that are filtered for GC content, palindromic sequences, alignment to host genomes, runs of identical bases and melting temperatures (lines 248-352). How was appropriate distance achieved?

We first generated all possible barcodes, applied the filtering criteria and then determined the Levenshtein distances. In Supplementary Figure 1 we show the distance between the 64 empirically tested barcodes. In Supplementary_data_file_1, we list 500 barcodes with the minimal Levenshtein distance to any other barcode (which stays above 14 for the whole barcode, and above 5 for the unique primer). If less than a million WISH-tags are used at a time, the likelihood of a compromised minimum distance for the entire 40 bp is very small. The list can be checked easily with the script we used to generate Supplementary Figure 1 (the link to the Zenodo site is provided in the manuscript).

Line 122: "...mice raised in a pathogen free environment..." Was it not a germ-free environment? The same mice are referred to later in the manuscript as "Germ-free (GF) mice" (line 193).

We have rephrased the text to clarify that the mice were raised in isolators in germ-free or Oligo-MM12 environment. Later, with the transfer to our experimental facility, these mice have been kept under pathogen-free conditions. We now refer to our standard protocol (Maier et al., 2014).

Line 123: "From now on, we will refer to this pooled DNA as BPM-DNA." It would be helpful to explain the reason for calling the pooled background DNA "BPM-DNA" – it only became apparent later that this must stand for "Bacterial – Plant – Mouse". Also, in the methods section there is a similar description of creating a mixed DNA pool, along with the statement, "This mix, from now on referred to as background DNA..." (line 376). Is this not the same thing? If they are please be consistent with the naming.

We adjusted the text and now define the term BPM-DNA when used the first time ("From now on, we will refer to this pooled DNA as BPM-DNA, which stands for bacteria, plant and mouse DNA"). The background DNA mentioned in the methods section indeed referred to the same BPM-DNA. The terms were now harmonized, and BPM-DNA is used throughout.

Line 185: Here the manuscript says that fecal pellets were collected for microbiota assessment, noting that "fecal *S. Tm* populations recapitulate the *S. Tm* population in the cecum". But the methods section on "DNA extraction mouse feces" describes overnight cultivation prior to genomic DNA extraction (lines 591-595) – does this recapitulate the cecum populations?

It is indeed critical that the relative proportions are reflected properly in the data. We have previously shown that that the cecum population is well reflected in fecal samples. We now refer to Meier et al., 2014, reference number 19. The corresponding sentence was adjusted to improve clarity.

Line 243: Here and elsewhere the reader is referred to Supplementary Figure 10 for details on experimental methods, but Supplementary Figure 10 displays Lipocalin-2 values. I believe the figure that should be referenced is Supplementary Figure 8.

Similarly, in the paragraph starting on line 244, readers are pointed to Supplementary Figure 13 for data

on colonization density in the context of the background community, which looks like it is actually in Supplementary Figure 11, and to Supplementary Figure 14q, which does not exist, for the statement “The carrying capacity remained the same as in the wild type Leaf257” which I don’t fully understand but I think is supposed to be represented in Supplementary Figure 12 (which does have a panel q). Overall, the overwhelming number of supplementary figures and their poor referencing in the text tend to detract from the message rather than enhance it.

We corrected the referencing of Supplementary Figures 8 and higher. We are sorry for the confusion.Line 282: The manuscript recommends the mBARq tool and references a submitted manuscript. I appreciate that this was provided to the reviewers, but maybe a pointer to a site where the software is available would be more helpful to readers particularly if the other manuscript isn't published yet?

The manuscript containing the links to the software was updated:
<https://doi.org/10.1101/2023.11.27.568830>. A link to the mBARq tool is provided therein.

Lines 410 and 415 (and others): One of the Salmonella strains is described as both "ATCC1428S" and "ATCC14028S" – I assume these are the same strain and only one of the names is correct? Also, on lines 471-475 it is noted that one barcoded strain of this genotype consistently failed to colonize and is assumed to "be afflicted by an off-target effect which reduces its fitness in the inflamed gut." But was this barcoded strain discarded or still used in the experiments?

We appreciate the thoroughness. Indeed, they are the same strain. We corrected the typo. Now all mentions display the correct "ATCC14028". The wild-type *S. Tm* strain that was mentioned in lines 471-475 was not used in any experiments sown in this study.

Line 522: "...samples were collected which were spotted on selective and non-selective R2A square plates to determine CFU and wild-type to barcoded strain ratio." How was this ratio determined?

The sentence was improved for clarity. The barcoded strain still harbours the antibiotic resistance used for the integration of the WISH-tag. The ratio was determined by assessing the CFU from non-selective with selective media. We now included this information.

Lines 532 and 537: "Mouse AW229 was removed from analysis because of clearing of all *S. Tm* from the gut for unknown reasons." And "Eight mice in the GF Early-Late, one of which was removed from the downstream analysis because of clearing of *S. tm*." Do these both refer to the same mouse that was removed? Also the second is an incomplete sentence.

They do refer to the same mouse. We adapted to the text to improve clarity. It now reads: "Eight mice in the GF Early-Late, one of which (AW229) was removed from the downstream analysis because of clearing of *S. Tm* from the gut".

Line 574: The manuscript says some plants were not harvested because of abnormal phenotypes, clarifying "meaning they were either very small.." but there is no "or" to continue this explanation.

We fixed the broken sentence. It now reads "... either very small or had deformed leaves ..."

Line 583: The entire plant was completely submerged in 600 uL buffer? Weren't some of these plants 7

or 14 days old?

Under the growth conditions, here calcined clay, the plants do not grow very large (similar to environmental conditions). Up to the second inoculation, when they were about 17 days old, they fitted into the volume. At the final harvest time, they needed to be compressed into the liquid by forcefully tapping the tube on the table.

Line 597: Why was 400 uL of Leaf257 culture added to “plant wash-off” DNA extractions?

Indeed, it was necessary to improve the extraction protocol due to the relatively low number of bacteria in each plant sample. We use the MasterPure DNA extraction kit, which relies on an isopropanol precipitation of the DNA in one of the steps. To ensure enough DNA was present for precipitation, we systematically added bacteria (untagged) to “bulk-up” the samples. The chromosomal DNA does not interfere with the amplification of the barcodes for sequencing. We adjusted the text to provide the rationale for the protocol we developed.

Line 628: The data availability statement lacks an accession number. Have the data been deposited?

We apologize for the oversight. The accession number PRJEB66333 was added to the manuscript.

Figure 1: It would be helpful if lengths of all parts of the tags we included in panel a. Panel C as well.

We double checked the proportionality of the sequence lengths relative to the indicated 40, 88 and 120 bp numbers. We would prefer not to include additional numbers, if possible. We provide the length of all sequence parts in the figure caption.

Figure 3: What is the “grey area” referred to in panel B?

The figure caption has been updated. Grey was referenced by mistake.

Figure 4: This legend points to Supplementary Figure 17 for tag abundance data, but that figure contains cloning procedures; I think Supplementary Figure 15 contains the relevant information.

The numbering of all Supplementary Figure 8 and higher were shifted by accident, we apologize. This is corrected in the revised version.

Supplementary Figure 15: the legend describes “...the right half with the grey background...” but there are grey panels interspersed with white panels and it’s not readily clear to me what is what.

We have updated the figure to be consistent with the figure caption.

Minor comments:

Italicization is inconsistent for terms like *in vitro*, *in situ*, *in planta*, etc. – either they should all be italicized or none.

We changed the terms to be consistently italicized.

Lines 147-148: It was hard to follow this sentence and understand that the three *Salmonella* strains are

ATCC14028S, SL1344 (SB300), both of which are commonly used in enteric bacteria research, and an avirulent mutant of SL1344.

We reformulated the sentence to enhance clarity:

“Three *Salmonella enterica* serovar Typhimurium (*S. Tm* from here on) strains, were barcoded using the lambda-red system in combination with pSIM for site directed integration^{44,45}. These strains were ATCC14028S⁴¹ and SL1344 (SB300)⁴², which are commonly used in the field of enteric bacteria research, as well as an avirulent mutant of SL1344 with deletions in $\Delta invG$ and $\Delta ssaV$.”

Line 315: Should be “priority effect” not “priority affect”.

We corrected the spelling error.

Line 327: Consider rephrasing “...can be used in specific but also across microbiomes...”

We followed the advice and rephrased the sentence for more clarity. It now reads: “To conclude, we have developed WISH-tags, novel genomic barcodes allowing their quantification with qPCR and NGS. After insertion into model and non-model bacteria, they can be applied to specific or also across multiple microbiomes.”

Line 488: There is an extra close parenthesis

We removed the extra parenthesis.

Line 544: Incomplete sentence

We rephrased the sentence: “After restreaking the WISH-tagged Leaf257 isogenic-lines from their individual glycerol-stocks on R2A+M plates, we grew them at 22°C for two days”

Reviewer #4 (Remarks to the Author):

This work by Daniel et al. introduces a new genome tagging system designed for the tracking the abundance of isogenic strains in complex microbial communities. The approach is straightforward and is compatible with qPCR and Illumina sequencing making it applicable in diverse scenarios. The authors validated the quantitative aspects of their approach, demonstrating its capability to span 5 orders of abundance magnitude. As a proof of concept, the approach was tested on a murine and plant system, revealing a priority effect in mice but not in the Arabidopsis phyllosphere.

The strengths of this study include the ability to discriminate, track, and characterize closely related strains of the same species within a complex native microbiota in a reliable, quantitative manner. The weaknesses of the study include a requirement for isolation of the strain under laboratory culture conditions, and low throughput tagging of these isolates. The demonstration experiments were performed under conditions of limited complexity, and the biological findings from these experiments are somewhat expected and do not advance our understanding of these host-microbiota ecosystems in a substantive manner. Overall, the contributions of this paper are more strongly oriented towards

29methods than towards biology.

Major points

1. Consider explaining in the Discussion the limitations on the species that can be tagged. For example, what are the implications of requiring laboratory growth? How many of the numerically most abundant native strains from each of these experimental systems are reliably cultivated and how representative

are the cultivated strains of each of these species? (*S. Typhimurium* and *E. coli* are not common commensals of *Mus musculus*.)

It is indeed an inherent problem to be able to cultivate a bacterial strain to introduce a tag. However, great progress has been made in culturing bacteria and making them available through public repositories (e.g. OligoMM, At-SPHERE). This provides a basis for testing strains for genomic integration across labs. With our study, we like to provide a resource for insertion sequences and strains that have already been tagged. We certainly hope that the number of tagged strains will grow upon time. We adjusted the discussion to make clear that the limitation is not the biological system but rather the strain accessibility. It now reads: “This demonstrates that the WISH-tag approach can be used in a wide range of genetically accessible microbial strains from various biological systems.”

2. How conserved is the *glmS* gene in the indigenous microbiota of these model systems? How important is this gene to the proposed strategy? How common are Tn7 transposase sites close to *glmS*? Of note, the authors report that several sites had to be tested in the *Rhizobium* strain of interest, before an appropriate one could be found. This suggests that the process of generating tagged strains may be quite laborious.

The *glmS* gene is widely conserved in Proteobacteria, which represent the dominant phylum in plant microbiomes. However, it is also present in Firmicutes/Bacteroidetes that are more abundant in gut systems. ([https://doi.org/10.1016/S0167-4838\(02\)00318-7](https://doi.org/10.1016/S0167-4838(02)00318-7)). The site is not a prerequisite for tagging. Any genomic site can be used, in principle, if fitness neutral. In this work, three independent integration methods are used. The Tn7 integrations are a convenient tool for integrations in many strains (<https://doi.org/10.1038/nprot.2006.24>). However, there are many alternative integration strategies that could be employed to integrate the WISH-tags.

The Tn7 transposon has coevolved with its host to integrate in a locus close to *glmS*, which is widespread. The transposase recognizes a region close to the end of the gene and integrates the entire transposon upstream of the gene, which has been shown not to cause downstream effects in various strains. However, not all *glmS* analogs are suitable targets for Tn7 transposition. We think this might have been the case for *Rhizobium* Leaf68. It harbours a *glmS* site, but integration using the Tn7 transposase remained unsuccessful, probably due to the recognition sequence being too different from the consensus sequence recognized by the Tn7 transposase. We had known from previous research that knock outs in this specific strain were possible using homologous recombination. We therefore chose to insert the tag in this strain at the *glmS* site, using homologous recombination. In short, integration of genetic elements into non model organisms can indeed be laborious, but once a suitable protocol is established, integrating additional barcodes can be relatively straightforward.

3. The discussion can benefit from addressing genetic changes that can in theory occur during the tagging process and/or after colonization. Consider sharing the number of generations that it takes to

tag a strain. This comment is related to what seem to be pre- and post- inoculation differences shown in Supp. Figure 9.

This raises an interesting and important point. We agree that mutations can occur during strain construction. In case of *S. Tm*, the WISH-tag was introduced into the wild type strain by the method of Datsenko and Wanner, as described in Materials and Methods. Afterwards the tag, which is linked with an ampicillin resistance cassette is transferred into a fresh *S. Tm* strain (*invGssaV* in this case) by P22 transduction. This classical method minimizes the chances that random genetic changes might occur during strain construction. Nevertheless, the strain is afterwards purified (to remove residual P22 phage)

by 3 re-streaks so that the stocked tagged strain will have undergone approximately 100 generations. For preparing the inoculum, we are growing up a single colony on plate and use it for inoculation and an overnight culture. This adds another approx. 50 generation and will inevitably result in an inoculum that has a few cells with random mutations in the genome. In our experience, the growth in LB does not lead to the selection of specific background mutations and our protocols are specifically designed to minimize their frequency. Furthermore, we are always testing 2 independent stocked WISH-tagged strains in our experiments. The absence of incongruent results thereby verifies that the phenotypes are not affected by random mutations which may have occurred during construction.

Within the mouse, strong selection for particular genotypes can inevitably occur. In our experience, the vast majority of mutations leads to attenuation. Due to the low frequency of such mutants in the inoculum, they should not affect the outcome of our in vivo colonization experiment. Only in the inflamed gut did we observe a few cases of positive selection in the past, i.e. of *hilD* or *tsr* mutants which are far fitter than the wild type (Diard et al., Nature 2013). However, in the present mouse colonization experiment, we can exclude this scenario, since we are working with an *invGssaV* mutant strain background, which does not trigger gut inflammation. The absence of gut inflammation is also verified by lipocalin-2 ELISA (Supplementary Figure 10).

Based on these considerations, we suspect that the decline of some tags in some mice is attributable to bottleneck effects during mouse gut colonization. Regardless, the use of 3 different WISH-tags per inoculum guarantees that we can still interpret our data with respect to the priority effect.

4. Further discussion would be useful about the advantages of this method over state-of-the-art strain-resolved metagenomic assembly methods.

The main advantages of this method over state-of-the-art strain-resolved metagenomic assembly methods are a relatively low cost coupled with high throughput, while being easy to analyse and interpret without an extensive bioinformatics background.

5. The authors should make clear what new information about priority effects was learned in these experiments that was either not known previously or could not have been learned using existing methods? How is this method superior to state-of-the-art strain-resolved metagenomic assembly methods?

Intra-strain specific priority effects were not addressed in the phyllosphere before. A metagenomic assembly methods would not be applicable to quantify barcoded strains. The price would be prohibitive. Only the amplification of the tag is necessary to address the biological questions addressed in our study.

Minor points

6. Consider specifying in that abstract that exactly two species were tested for a priority effect. As

written, it is not clear if Salmonella and Sphingomonas were the only species assessed.

We added this information to improve the clarity of the abstract. It now reads “We then investigated intra-strain priority effects using one species of isogenic barcoded bacteria for each, the murine gut and the *Arabidopsis* phyllosphere, both with and without microbiota context.”

7. Line 20. The claim of no background DNA amplification should be modified. More precisely, there was no background DNA detected above the limits of detection. The claim should be revised to reflect this,

and ideally state the limits of detection. Similarly in line 98 and elsewhere, the claim that background DNA was “excluded” should be revised.

We added “detectable” to the respective sentence and double-checked phrasing throughout. As for our claim that background DNA was “excluded”, we modified it to “not detectable”

8. Line 47. The claim that barcodes are ideal for tracing population composition is over-stated. They have advantages and disadvantages. Among the latter, they fail to allow for monitoring extrachromosomal elements, and other forms of genome evolution.

They are not intended nor suitable to trace extrachromosomal elements or genome evolution, and we make no such claim. They are, however, suitable for lineage tracing and by extension allow to inexpensively monitor the fitness impact of changes in the genome on the relative fitness with competing lines.

9. Line ~50. Contrast with in vivo murine work like Vasquez et al. Cell Host Microbe 2021, 10.1016/j.chom.2021.08.003.

We now refer to the study, as suggested.

10. Line 57. In theory, no insertion into a genome is perfectly neutral. What the authors may mean by neutral is the absence of a detected fitness effect. As with any negative finding, it may be important to qualify the finding with the sensitivity of the measurement method.

So far, no impact of Tn7 integration has been reported. We have adjusted the text. It now reads: “Together with the insertion of the barcodes at neutral sites, this allows the generation of independently traceable isogenic strains that differ in the 40 bp of their barcode. The application of independent tags is a common practice to mitigate the impact of spontaneous mutations. “

11. Line 63. Gnotobiotic animal models differ from naturally colonized animals of the same host lineage in more than the absence of microbiota. Having been born to a germ-free dam/mother, they fail to complete post-natal tissue and organ differentiation. They have very different host physiology at birth, and even after mono- or more complex colonization, host development follows a different trajectory. To the degree that host biology and immunology control microbiota colonization, gnotobiotic animals provide a somewhat aberrant view of microbiota assembly.

We agree, the influence of the microbiome on host development is a known factor. However, the Oligo-MM12 community is consistent in colonizing the gut and transfer over generations is stable (<https://doi.org/10.1038/nmicrobiol.2016.215>). Regarding the comments about the physiology, this has

been extensively studied in a recent publication (<https://doi.org/10.1016/j.chom.2022.09.011>, especially Figure 6 of the paper), where the authors show that the changes in Oligo-MM12 are an intermediate between SPF and GF animals. Therefore, by choosing the Oligo-MM12 mice, we retain the relative ease of control over the input and community, while moving away from potential drawbacks of GF animals.

12. Line 99. To avoid confusion please correct “the only difference between the tags” -> “tags differed in a 40bp region” or similar.We followed the reviewer's suggestion to rephrase the sentence. It now reads: "The newly developed WISH-tags differed only in their core, the 40 bp of the unique barcode region, which ensures sufficient distinctiveness between any two tags."

13. Figure 2a. Did you add ~100k molecules per tag? If you did, you should specify that in the legend. In the current wording, you specify only "equal concentrations". Also – please state that the dashed horizontal line is the detection limit, and either define it here in legends or refer to a definition elsewhere (e.g., in supp. fig. 2).

The *E. coli* harbouring the WISH-tags on plasmids were grown in parallel and under the same conditions. We then mixed them at equal concentrations (as described in the methods section). Calculating the copy number using the DNA concentrations measures in the template we arrive at ca. 123645 copies per barcode. Therefore, we amended the figure caption to reflect that.

We added a description of the detection limit: "The dashed line represents the limit of detection, set at a Ct value of 32."

14. Figure 2b. Please define "normalized counts".

A definition can be found in the methods, which we have amended for improved clarity: "These results were normalized to the total number of sequences per library, and used to create Figures 2 b, c and d."

15. Figure 2b. Why are there so many outliers at dilution 7? Getting suddenly 1000 reads when you added close to nothing seems strange.

We assume the question was directed at Figure 2c. As we mention in the manuscript, we were puzzled by these reads as well. All eight mixes were generated from the same material according to Supplementary Figure 3. We have excluded a lot of sources for these reads that are close to the detection limit but could not pinpoint the cause.

16. Figure 2c. Did you choose not to show dilutions 6 and 7? If so, please state such and consider expanding in the legends or main text on NGS issues when the number of molecules is very small or zero.

We assume the question was directed at Figure 2d. We now state the exclusion of dilutions 6 and 7 in the figure caption. When the overall number of template molecules is low, each sample is more strongly amplified during library preparation, giving rise to more amplification artefacts.

17. Line 149. "were barcoded" – with how many barcodes? Reading further it seems that "six barcodes each" on line 151 may refer to all strains. Please clarify.

We rephrased the statement to improve clarity. It now reads: “We barcoded the mouse gut commensal *E. coli* 8178⁴³. In addition, we tagged six representative species of the At-LSPHERE with six barcodes, unique to the species, each.”

18. Line 153. Not clear for an outsider why you need a different integration system. If possible, explain briefly. Also explain here or in the discussion which bacteria are suitable for this approach, and which are not. (see Comment #)We have added a sentence to clarify that the WISH-tags are not dependent on a specific system for genomic integration: “Since the WISH-tags are independent of the integration method, we choose the most readily available one for each strain.”

As to the question, which bacteria are suitable, WISH-tags can be applied in and genomically tractable organism (line 148).

19. Line 158. Please add a separate statement or sentence for all supplementary figures 5 to 9.

As recommended, we added a short description for each supplementary Figure: “Supplementary Figure 5: Validation of mouse gut associated strains. Supplementary Figure 6: *In planta* validation of *Sphingomonas* Leaf257. Supplementary Figure 7: *in vitro* validation of *At*-LSPHERE species.” Supplementary Figures 8 and 9 were mistakenly listed here.

20. Supp. Figure 9. Looks like the tagged strains might not be genetically identical. For example, WISH-25 is expanding while WISH-27 is declining in the 4 mice in panel a, oligo-MM. Would you assess these differences to be beyond neutral drift? Perhaps add comment in discussion about the possibility of *in vivo* evolution or other sources of genetic variance such as pre-inoculation differences.

There are multiple possible explanations for this population dynamic. Since it is not always the same barcoded line, we speculate that this might be caused by neutral drift that is synchronized between the mice, as they are co-housed and no measures to prevent coprophagy were taken.

21. Supp. Figure 11. “The y-axis shows the time and the x-axis the CFU per plant.” The authors have mixed up the x- and y-axes in this statement. All legends to Supplementary Figures should be carefully checked for wording and errors.

We appreciate the thoroughness of the revision and addressed the mistake. We also rechecked the remaining figure legends.

22. Around line 190. Did you use shotgun or qPCR here? Please be explicit about the approach taken.

We added the information: “The barcodes were quantified by illumina sequencing.”

23. Line 206 and Figure 3b. I can’t clearly see a pink line (I am slightly color blind), a gray area, or arrows. Please enhance figure quality and perhaps make colors more pronounced if needed.

Thank you. We adjusted the colors and figure legend.

24. Figure 3c. I didn’t see a reference from the main text to Figure 3c.

We included a reference to Figure 3c in the revised manuscript.

25. Line 250. Is this reference to “Supp. Figure 14q” intended to refer to “Supp. Figure 12q”?

There was a legacy mistake from an older version of the manuscript. All figures past Supplementary Figure 7 were shifted by two. We apologize and corrected the numbering.

26. Line 254. “Supplementary Figure 16be” seems to be a typo.

We intended to refer to Supplementary Figure 14b and e. We rephrased it for clarity: “The focal strain reached the same carrying capacity, regardless of whether it was inoculated at 0 dpi or 7 dpi and the mock inoculation at 7 dpi did not reveal any reduction in bacterial populations compared to untreated plants (Supplementary Figure 12k, Supplementary Figure 14b and e), indicating that the number of bacteria removed by the second inoculation was below our detection limit.”

27. Line 261. “Supplementary Figure 17” may also be a typo.

We intended to refer to Supplementary Figure 15. We have corrected this.

28. Line 407. Allowing up to 5 errors seems unnecessary since a typical sequencing error rate is only 0.5% per bp. Do you think this might explain the false positives in Figure 2c, dilution 7?

When we permitted up to 5 errors, the number of reads barely changed, so we do not expect this to have a large impact. The false positives are probably not caused by this allowance, as they only appeared in two of the eight libraries.

29. Line 424. Please refer to Supp. Figure 17 from the main text. Could be informative for non-experts to hear about the integration process in a single sentence.

We are now referring the reader to the methods section, where the cloning is explained in more detail.

Decision Letter, first revision:

Message: Our ref: NMICROBIOL-23082038B

18th December 2023

Dear Julia,

Thank you for submitting your revised manuscript "Wild-type Isogenic Standardized Hybrid (WISH)-tags for assessing population dynamics within microbiomes" (NMICROBIOL-23082038B). It has now been assessed editorially and we find that the paper has improved in revision, and therefore we'll be happy in principle to publish it in Nature Microbiology, pending minor revisions to comply with our editorial and formatting guidelines.

Please can you let us know the status of your mBARq paper i.e. at what stage of review?

Thank you again for your interest in Nature Microbiology Please do not hesitate to contact me if you have any questions.

Sincerely,

Final Decision Letter:

Message: 9th February 2024

Dear Julia,

I am pleased to accept your Resource "Assessing microbiome population dynamics using wild-type isogenic standardized hybrid (WISH)-tags" for publication in Nature Microbiology. Thank you for having chosen to submit your work to us and many congratulations.

2You may wish to make your media relations office aware of your accepted publication, in case they consider it appropriate to organize some internal or external publicity. Once your paper has been scheduled you will receive an email confirming the publication details. This is normally 3-4 working days in advance of publication. If you need additional notice of the date and time of publication, please let the production team know when you receive the proof of your article to ensure there is sufficient time to coordinate. Further information on our embargo policies can be found here:

<https://www.nature.com/authors/policies/embargo.html>

Please note that *Nature Microbiology* is a Transformative Journal (TJ). Authors may publish their research with us through the traditional subscription access route or make their paper immediately open access through payment of an article-processing charge (APC). Authors will not be required to make a final decision about access to their article until it has been accepted. Find out more about Transformative Journals

With kind regards,